# Qualitative and Quantitative Detection of Potentially Virulent *Vibrio parahaemolyticus* in Drinking Water and Commonly Consumed Aquatic Products by Loop-Mediated Isothermal Amplification

**DOI:** 10.3390/pathogens11010010

**Published:** 2021-12-22

**Authors:** Zhengke Shen, Yue Liu, Lanming Chen

**Affiliations:** Key Laboratory of Quality and Safety Risk Assessment for Aquatic Products on Storage and Preservation (Shanghai), Ministry of Agriculture and Rural Affairs of the People’s Republic of China, College of Food Science and Technology, Shanghai Ocean University, Shanghai 201306, China; M190300756@st.shou.edu.cn (Z.S.); M190300758@st.shou.edu.cn (Y.L.)

**Keywords:** *V. parahaemolyticus*, foodborne pathogen, molecular diagnosis, LAMP, drinking water, aquatic product

## Abstract

*Vibrio parah**aemolyticus* can cause acute gastroenteritis, wound infection, and septicemia in humans. In this study, a simple, specific, and user-friendly diagnostic tool was developed for the first time for the qualitative and quantitative detection of toxins and infection process-associated genes *opaR*, *vpadF*, *tlh,* and *ureC* in *V. parahaemolyticus* using the loop-mediated isothermal amplification (LAMP) technique. Three pairs of specific inner, outer, and loop primers were designed for targeting each of these genes, and the results showed no cross-reaction with the other common *Vibrios* and non-*Vibrios* pathogenic bacteria. Positive results in the one-step LAMP reaction (at 65 °C for 45 min) were identified by a change to light green and the emission of bright green fluorescence under visible light and UV light (302 nm), respectively. The lowest limit of detection (LOD) for the target genes ranged from 1.46 × 10^−5^ to 1.85 × 10^−3^ ng/reaction (25 µL) for the genomic DNA, and from 1.03 × 10^−2^ to 1.73 × 10^0^ CFU/reaction (25 µL) for the cell culture of *V. parahaemolyticus*. The usefulness of the developed method was demonstrated by the fact that the bacterium could be detected in water from various sources and commonly consumed aquatic product samples. The presence of *opaR* and *tlh* genes in the *Parabramis pekinensis* intestine indicated a risk of potentially virulent *V. parahaemolyticus* in the fish.

## 1. Introduction

*V. parahaemolyticus* is a Gram-negative bacterium that can cause acute gastroenteritis, wound infection, and septicemia in humans [1]. The bacterium inhabits estuarine and marine environments worldwide, and is also frequently detected in aquatic products [2]. Clinical *V. parahaemolyticus* isolates produce two major toxins, thermostable direct hemolysin (TDH) and TDH-related hemolysins (TRH), both of which cause hemolysis and cytotoxicity of the host cells [3]. Their encoding genes *trh* and *tdh*, sharing approximately 70% homology, are molecular markers for the diagnosis of virulent *V. parahaemolyticus* isolates [4].

Previous studies have also revealed very important virulence-associated genes in *V. parahaemolyticus*; for example, an *ureC* gene encodes urease subunit alpha, which is known to be associated with enterotoxicity, a reasonably good clinical diagnostic marker for *trh*-positive *V. parahaemolyticus* isolates [5]. A *tlh* gene encodes a thermolabile hemolysin (TLH) that is present in pathogenic and non-pathogenic *V. parahaemolyticus* isolates [6]. The TLH is one of the phospholipases that can hydrolyze glycerophospholipids, the major component of the eukaryotic cell membrane, and disrupt the host cells. Therefore, it is a key virulence factor in many pathogenic bacteria [7]. Pathogen adhesion subverts the host actin cytoskeleton and triggers cellular signaling pathways to facilitate subsequent pathogen invasion [8]; for example, a *vpadF* gene encodes an adhesion factor that enables *V. parahaemolyticus* to interact with type I collagen and mediate a type III secretion system on chromosome 2 (T3SS2)-dependent host cell invasion [9]. Additionally, an *opaR* gene encodes a master quorum sensing (QS) regulator of *V. parahaemolyticus*, and regulates the transcription of many genes involved in virulence, motility, and biofilm formation [10]. It is regarded as an attractive target to combat bacterial pathogenicity, with the potential to be used as a vaccine candidate [8]. Thus, the diagnosis of these virulence-associated genes in *V. parahaemolyticus* is imperative for food safety control and human health.

Many methods have been developed to detect pathogenic bacteria, based on microbiology, biochemistry, immunology, spectroscopy, and molecular biology technology [11]. These methods are usually laborious, time consuming, or require costly and bulky equipment [12]. Conversely, the loop-mediated isothermal amplification (LAMP) technique, originally developed by Notomi et al. [13,14], can amplify target genes at a constant temperature with a one-step reaction, exclusion of a thermal cycler that is needed by the standard polymerase chain reaction (PCR), and reverse transcription-PCR (RT-PCR) assays [15]. Successful amplification in LAMP reactions can be directly visualized via a variety of visual indicators, such as hydroxynaphthol blue, phenol red dye, hydroxy naphthol blue and leuco crystal violet, and the nucleic acid dyes SYBR Green I and SYTO 9 [16,17,18,19,20]. Recently, a MnCl_2_–calcein dye has been applied in LAMP to circumvent the instability problem of other dyes [21]. Calcein can combine with divalent metal ions (such as Ca^2+^ and Mg^2+^) to form complexes, and produces strong fluorescence [18]. In the LAMP reaction system, if no target gene amplification occurs, calcein binds with Mn^2+^ to cause fluorescence quenching, and the reaction is orange–yellow. In contrast, when the target sequence is amplified, the Mn^2+^ bound to calcein is deprived by the newly generated pyrophosphate ions, and calcein binds to the residual Mg^2+^ in the reaction system. Consequently, the fluorescence is enhanced and the reaction is green [22]. LAMP technology has been applied in the detection of human infectious diseases, such as severe acute respiratory syndrome (SARS) [23], avian influenza virus (AIV) [24], hemagglutinin 1 neuraminidase 1 (H1N1) [25], coronavirus disease 2019 (COVID-19) [26], as well as common foodborne pathogenic bacteria, e.g., *Vibrio cholerae* [27,28], *Staphylococcus aureus* [29], and *Salmonella* species [30]. Studies have been conducted to test the toxin-associated genes of *V. parahaemolyticus* by LAMP, such as *tdh*, *trh*, *toxR*, and *groEL* [3,31,32,33]; nevertheless, current literature in this field for the *opaR*, *tlh*, *vpadF* and *ureC* genes is rare.

In our previous studies, the LAMP reaction system was well established in our research group; for example, an sssvLAMP method was developed to detect the causative agent of cholera, the *V. cholerae*-specific gene *lolb*, the toxin genes *ctxA* and *tcpA*, and the virulence-associated genes *hapA*, *mshA*, *pilA*, *tlh*, *nanH,* and *cri* [27,28]. In this study, the qualitative and quantitative detection of the very important virulence-related genes *opaR*, *tlh*, *vpadF,* and *ureC* of *V**. parahaemolyticus* in drinking water and commonly consumed aquatic products was developed for the first time using the LAMP technique. The objectives of this study were as follows: (1) to design three pairs of specific primers targeting each of the *opaR*, *tlh*, *ureC*, and *vpadF* genes of *V. parahaemolyticus*; (2) to determine the specificity and sensitivity of the LAMP method for the detection of cell culture and genomic DNA, as well as spiked samples of *V. parahaemolyticus*, and compare these with the standard PCR assay; (3) to rapidly screen potentially virulent *V. parahaemolyticus* in drinking water and commonly consumed fish, shrimp, and shellfish specimens by the LAMP method. The results of this study provide a simple, specific, and user-friendly molecular diagnostic tool for early diagnosis, particularly for the large-scale screening of drinking water and aquatic products contaminated by *V. parahaemolyticus*, a leading sea foodborne pathogen worldwide.

## 2. Results

### 2.1. Specificity of the LAMP Method

A total of 50 bacterial strains were employed in the exclusivity tests, including 7 species of *Vibrios* (*n* = 16 strains) and 20 species of non-*Vibrios* (*n* = 34 strains) (Appendix A). Some common pathogenic bacteria were included, e.g., *Vibrio alginolyticus*, *Vibrio fluvialis*, *Vibrio harvey**i*, and *Vibrio vulnificus*, as well as *Aeromonas hydrophila*, *Enterobacter sakazakii*, *Klebsiella oxytoca*, *Klebsiella pneumonia**e*, *Listeria monocytogenes*, *Pseudomonas aeruginosa*, *Enterobacter cloacae*, and *Staphylococcus aureus* (Appendix A). Pathogenic *V. parahaemolyticus* ATCC17802 (*opaR^+^*/*vpadF^+^*/*tlh^+^*/*ureC^+^*) was used as a positive control strain. For the target gene *opaR*, the LAMP reaction tube containing the genomic DNA sample extracted from *V. parahaemolyticus* ATCC17802 (*opaR^+^*) showed a color change from orange to light green under visible light, after being reacted at 65 °C for 45 min, whereas the other 50 tubes containing each of the DNA templates extracted from the 50 bacterial strains showed the original color of orange (Figure 1A). The positive reaction tube was also observed under ultraviolet (UV) light (302 nm), which emitted bright green fluorescence, whereas the 50 negative reaction tubes had no fluorescence (Figure 1B).

Similarly, for the target genes *tlh*, *ureC*, and *vpadF* in the exclusivity tests, only the LAMP reaction tubes containing the genomic DNA of *V. parahaemolyticus* ATCC17802 (*vpadF^+^*/*tlh^+^*/*ureC^+^*) showed the color change and emitted bright green fluorescence, while the other 50 tubes containing the other DNA templates were orange with no emission of fluorescence (figures not shown).

These results were confirmed by standard agarose gel electrophoresis analyses, in which the LAMP products from the positive reaction tube formed characteristic ladder-like DNA patterns [22], while those from the negative reaction tubes showed no DNA bands. Taken together, the LAMP method was highly specific to target each of the *opaR*, *tlh*, *ureC*, and *vpadF* genes of *V. parahaemolyticus* (Table 1), and no cross-reaction was observed with the other 7 species of *Vibrios* and 20 species of non-*Vibrios* strains tested in this study.

### 2.2. Sensitivity of the LAMP Method

#### 2.2.1. For the Detection of Cell Culture of *V. parahaemolyticus*

A total of 51 *V. parahaemolyticus* strains were employed in the inclusivity tests, and the results are presented in Table 2, Table 3 and Table 4. For the target gene *opaR*, for example, serial dilutions of *V. parahaemolyticus* B4-13 cell culture (1.32 × 10^9^ to 1.32 × 10^0^ CFU/mL) were added into the LAMP reaction tubes. After being reacted at 65 °C for 45 min, the limit of detection (LOD) was observed in the tube containing 4.40 CFU/reaction (25 µL) of *V. parahaemolyticus* B4-13 cells, which changed color to light green and emitted bright green fluorescence under visible light and UV light, respectively (Figure 2A(r1 and r2)). The LOD tube also formed characteristic ladder-like DNA patterns in the agarose gel electrophoresis analyses (Figure 2A(r3)). Similarly, for the detection of the *V. parahaemolyticus* B11-3 cell culture (7.00 × 10^8^–7.00 × 10^0^ CFU/mL), the observed LOD was 2.33 × 10^1^ CFU/reaction, while for the *V. parahaemolyticus* N3-3 cell culture (9.10 × 10^7^–9.10 × 10^0^ CFU/mL), the LOD was 3.03 × 10^−1^ CFU/reaction (Figure 2B,C). Additionally, the cell cultures of the other 47 *V. parahaemolyticus* strains (*opaR^+^*) were all tested in the inclusivity tests. For the target gene *opaR*, the observed LOD values of the LAMP method ranged from 1.03 × 10^−2^ to 7.13 × 10^3^ CFU/reaction for the detection of the cell cultures of the 50 *V. parahaemolyticus* strains (Table 2).

For the target gene *vpadF*, the cell culture of 39 *V. parahaemolyticus* strains (*vpadF^+^*) was examined in the inclusivity tests (Table 3); for example, serial dilutions of *V. parahaemolyticu*s B7-16 (2.34 × 10^8^ to 2.34× 10^0^ CFU/mL), B9-42 (5.20 × 10^7^ to 5.20× 10^0^ CFU/mL), and N4-46 (8.60 × 10^7^ to 8.60 × 10^0^ CFU/mL) were examined by the LAMP method. The results showed that their LOD tubes contained 7.80 × 10^1^ CFU/reaction, 1.73 CFU/reaction, and 2.87 × 10^2^ CFU/reaction of *V. parahaemolyticus*, respectively (figures not shown). Similarly, each of the other 36 *V. parahaemolyticus* strains (*vpadF^+^*) were all tested in the inclusivity tests. The results indicated that for the target gene *vpadF*, the LODs of the LAMP method ranged from 1.73 × 10^0^ to 8.63 × 10^3^ CFU/reaction (Table 3).

For the target gene *tlh*, the cell culture of 50 *V. parahaemolyticus* strains (*tlh^+^*) was tested in the inclusivity tests. Serial dilutions of their cell culture ranged from 2.14 × 10^9^ to 1.02 × 10^0^ CFU/mL. The results showed that the LODs of the LAMP method targeting the *tlh* gene ranged from 1.37 × 10^0^ to 9.00 × 10^3^ CFU/reaction (Table 4).

For the target gene *ureC*, serial dilutions of *V. parahaemolyticus* ATCC17802 (*ureC ^+^*, 1.32 × 10^8^ to 1.32 × 10^0^) were tested, and the observed LOD was 4.40 × 10^−1^ CFU/reaction by the LAMP method (Table 4, figures not shown).

Additionally, the target genes amplified from representative *V. parahaemolyticus* strains were confirmed by PCR and DNA sequencing analyses. The resulting sequences were deposited in GenBank under the accession numbers listed in Appendix A.

Taken together, approximately 41.2% of the *V. parahaemolyticus* strains (21 of the 51 strains) could be detected in less than 10 CFU/reaction (25 µL) by the LAMP method developed in this study, and the average detection time was 1.5 h, which highlighted the high sensitivity of the LAMP method for the detection of a cell culture of *V. parahaemolyticus*.

#### 2.2.2. For the Detection of Genomic DNA of *V. parahaemolyticus*

The sensitivity of the LAMP method for the detection of the genomic DNA of the 50 *V. parahaemolyticus* strains was also determined (Table 2, Table 3 and Table 4). For the target gene *opaR*, genomic DNA samples extracted from each of the 50 *V. parahaemolyticus* strains (*opaR*^+^) were serially diluted with concentrations ranging from 6.58 × 10^−6^ to 4.95 × 10^2^ ng/µL, and examined by the LAMP method. To conduct this sensitivity test, genomic DNA dilutions of *V. parahaemolyticus* L5-1 (2.10 × 10^2^ to 2.10 × 10^−5^ ng/µL) were added into LAMP reaction tubes. After being reacted at 65 °C for 45 min, eight tubes had positive reactions, showing a light green color, bright green fluorescence, and characteristic ladder-like DNA patterns (Figure 3A). The LOD tube contained 4.21 × 10^−5^ ng/reaction (25 µL) of genomic DNA. Similarly, genomic DNA dilutions of each of the other 49 *V. parahaemolyticus* strains were examined by the LAMP method. The results indicated that the LODs targeting the *opaR* gene ranged from 1.46 × 10^−5^ to 1.85 × 10^0^ ng/reaction of genomic DNA of *V. parahaemolyticus* using the LAMP method (Table 2).

Similarly, for the target gene *vpadF*, genomic DNA dilutions (6.58 × 10^−6^ to 4.95 × 10^2^ ng/µL) of each of the 39 *V. parahaemolyticus* strains (*vpadF^+^*) were tested in the LAMP tubes. The results showed that the LODs targeting the *vpadF* gene ranged from 1.85 × 10^−4^ to 4.30 × 10^−1^ ng/reaction using the LAMP method (Table 3, Figure 3B).

For the target gene *tlh*, genomic DNA dilutions of each of the 50 *V. parahaemolyticus* strains (*tlh*^+^) were tested in the LAMP tubes. The results showed that the LODs of the LAMP method for the *vpadF* gene ranged from 1.85 × 10^−4^ to 3.35 × 10^1^ ng/reaction (Table 4, Figure 3C).

For the target gene *ureC*, genomic DNA dilutions (9.26 × 10^1^ to 9.26 × 10^−6^ ng/μL) of *V. parahaemolyticus* ATCC17802 (*ureC^+^*) were examined by the LAMP method. The LOD tube contained 1.85 × 10^−3^ ng/reaction of genomic DNA for the *ureC* gene (Table 4, figures not shown).

Taken together, approximately 76.5% of the genomic DNA samples from all the *V. parahaemolyticus* strains (39 of the 51 strains) could be detected at less than 10 pg/reaction (25 µL) by the LAMP method developed in this study, which demonstrated the high sensitivity of the LAMP method for the detection of the genomic DNA of *V. parahaemolyticus*.

### 2.3. Sensitivity Comparison of the LAMP Method with the Standard PCR Assay

#### 2.3.1. For the Detection of Cell Culture of *V. parahaemolyticus*

To compare the sensitivity of the LAMP method with the standard PCR assay, serial dilutions of each of the 50 *V. parahaemolyticus* cell cultures (2.14 × 10^9^ to 1.02 × 10^0^ CFU/mL) were examined by the PCR assay. For the target gene *opaR*, the observed LOD values of the PCR assay ranged from 7.13 × 10^6^ to 1.37 × 10^3^ CFU/reaction via the routine agarose gel electrophoresis analysis (Table 2); for example, when serial dilutions of *V. parahaemolyticus* B4-13 cell culture were tested, the observed LOD of the PCR assay was 4.40 × 10^4^ CFU/reaction (Figure 2D), which was 1.00 × 10^4^-fold lower than that of the LAMP method (4.40 CFU/reaction) (Figure 2A(r4)).

Similarly, for the target gene *vpadF*, the cell culture of the 39 *V. parahaemolyticus* strains (*vpadF^+^*, 2.14 × 10^9^ to 1.02 × 10^0^ CFU/mL) was tested by the PCR assay. The resulting LODs of the PCR assay were 1.73 × 10^2^ to 8.63 × 10^5^ CFU/reaction (Table 3). Serial dilutions of *V. parahaemolyticus* B7-16, B9-42, and N4-46 were also examined by the PCR assay. The results showed that their LODs were 7.80 × 10^4^, 1.73 × 10^2^, and 2.87 × 10^4^, which were 1000-, 100-, and 100-fold lower than those obtained by the LAMP method (7.80 × 10^1^ CFU/reaction, 1.73 CFU/reaction, and 2.87 × 10^2^ CFU/reaction), respectively (Table 3, figures not shown).

For the target gene *tlh*, the cell culture of the 50 *V. parahaemolyticus* strains (*tlh^+^*) was tested by the PCR assay, and the observed LODs were recorded to range from 7.90 × 10^1^ to 9.20 × 10^6^ CFU/reaction. The LAMP method was 1.00 × 10^1^- to 1.00 × 10^4^-fold more sensitive than the PCR assay (Table 4).

For the target gene *ureC*, serial dilutions of *V. parahaemolyticus* ATCC17802 cell culture (*ureC ^+^*) were examined by the PCR assay. The LOD of the PCR assay was 4.40 × 10^0^ CFU/reaction, which was 10-fold less sensitive than the LAMP method (4.40 × 10^−1^ CFU/reaction; Table 4, figures not shown).

These results demonstrated that the lowest LODs, obtained using the PCR assay, targeting the *opaR*, *tlh*, *ureC*, and *vpadF* genes in the cell culture of *V. parahaemolyticus* strains, were 1.00 × 10^1^- to 1.00 × 10^7^-fold lower than those obtained using the PCR assay.

#### 2.3.2. For the Detection of Genomic DNA of *V. parahaemolyticus*

Genomic DNA dilutions of each of the 50 *V. parahaemolyticus* strains (6.58 × 10^−6^ to 4.95 × 10^2^ ng/µL) were also examined by the PCR assay, and the resulting data are presented in Table 2, Table 3 and Table 4. For the target gene *opaR*, the observed LODs for the detection of the genomic DNA of the 50 *V. parahaemolyticus* strains (*opaR^+^*) ranged from 1.96 × 10^−2^ to 3.79 × 10^2^ ng/reaction using the PCR assay, which was 1.00 × 10^1^- to 1.00 × 10^6^-fold lower than those obtained using the LAMP method (Table 2, Figure 3).

Similarly, for the target gene *vpadF*, the LODs for the detection of the genomic DNA of the 39 *V. parahaemolyticus* strains (*vpadF^+^*) were 2.68 × 10^−1^ to 4.16 × 10^2^ ng/reaction using the PCR assay, which was 1.00 × 10^1^- to 1.00 × 10^4^-fold lower than those obtained using the LAMP method (Table 3, Figure 3).

For the target gene *tlh*, the LODs for the detection of the genomic DNA of the 50 *V. parahaemolyticus* strains (*tlh^+^*) were 3.05 × 10^−2^ to 6.04 × 10^2^ ng/reaction using the PCR assay, which was 1.00 × 10^1^- to 1.00 × 10^4^-fold lower than those obtained using the LAMP method (Table 4, Figure 3).

For the target gene *ureC*, the observed LOD for the detection of the genomic DNA of *V. parahaemolyticus* ATCC17802 (*ureC*^+^) was 1.85 × 10^−1^ ng/reaction using the PCR assay, which was 100-fold lower than that obtained using the LAMP method (Table 4, figures not shown).

These results demonstrated that the sensitivity of the LAMP method was 1.00 × 10^1^- to 1.00 × 10^6^-fold higher than that of the routine PCR assay for the detection of the genomic DNA of *V. parahaemolyticus* strains.

### 2.4. Sensitivity of the LAMP Method for the Detection of Spiked Fish, Shrimp and Shellfish Samples

Cell cultures of the *V. parahaemolyticus* strains ATCC17802 (*opaR*^+^/*vpadF*^+^/*tlh*^+^/*ureC*^+^) and N7-19 (*opaR*^+^/*vpadF*^+^/*tlh*^+^/*ureC*^-^) were individually spiked into each of six species of commonly consumed aquatic animal samples, including the following four species of fish: *Aristichthys nobilis*, *Carassius auratus*, *Ctenopharyngodon idella*, and *Parabramis pekinensis*; the following species of shrimp: *Litopenaeus vannamei*; the following species of shellfish: *Mytilus edulis*. The sensitivity of the LAMP method was determined for each of the target genes, and the resulting data are presented in Table 5.

When the cell culture of *V. parahaemolyticus* N7-19 (2.96 × 10^8^ to 2.96 CFU/mL) was spiked into the samples, for the target gene *opaR*, the resulting LODs were recorded to range from 9.87 × 10^0^ to 9.87 × 10^2^ CFU/reaction for the spiked fish; 9.87 × 10^−2^ CFU/reaction for the spiked *L. vannamei**;* 9.87 × 10^2^ CFU/reaction for the spiked *M. edulis* samples. For the target gene *vpadF*, the LOD values were 9.87 × 10^−1^ to 9.87 × 10^2^ CFU/reaction for the spiked fish; 9.87 × 10^0^ CFU/reaction for the spiked shrimp; 9.87 × 10^2^ CFU/reaction for the spiked shellfish. For the target gene *tlh*, the LOD values of the LAMP method ranged from 9.87 × 10^2^ to 9.87 × 10^3^ CFU/reaction for the spiked fish; 9.87 × 10^2^ CFU/reaction for the spiked shrimp; 9.87 × 10^4^ CFU/reaction for the spiked shellfish samples (Table 5, Figure 4).

Similarly, when the cell culture of *V. parahaemolyticus* ATCC17802 (2.75 × 10^9^ to 2.75 CFU/mL) was spiked into the aquatic product samples, for the target gene *ureC*, the observed LODs by the LAMP method ranged from 9.17 × 10^3^ to 9.17 × 10^2^ CFU/reaction for the spiked *A. nobilis*, *C. auratus*, *C. idella*, and *P. pekinensis*; 9.17 × 10^2^ CFU/reaction for the spiked *L. vannamei*; 9.17 × 10^1^ CFU/reaction for the spiked *M. edulis* samples (Table 5, Figure 4).

### 2.5. Sensitivity Comparison of the LAMP Method with the Standard PCR Assay for the Detection of Spiked Aquatic Product Samples

The sensitivity of the detection of the spiked aquatic product samples with a cell culture of *V. parahaemolyticus* ATCC17802 (*opaR*^+^/*vpadF*^+^/*tlh*^+^/*ureC*^+^) and N7-19 (*opaR*^+^/*vpadF*^+^/*tlh*^+^/*ureC*^−^) was also determined by the standard PCR assay (Table 5); for example, when *V. parahaemolyticus* N7-19 was spiked into the samples, for the target gene *opaR*, the observed LODs using the PCR assay were 9.87 × 10^1^ to 9.87 × 10^4^ CFU/reaction for the spiked *A. nobilis*, *C. auratus*, *C. idella*, and *P. pekinensis*; 9.87 × 10^3^ CFU/reaction for the spiked *L. vannamei*; 9.87 × 10^3^ CFU/reaction for the spiked *M. edulis* samples. Similarly, for the target gene *vpadF*, the observed LODs using the PCR assay were 9.87 × 10^2^ to 9.87 × 10^4^ CFU/reaction for the spiked fish; 9.87 × 10^3^ CFU/reaction for the spiked shrimp; 9.87 × 10^4^ CFU/reaction for the spiked shellfish samples. For the target gene *tlh*, the observed LODs using the PCR assay were 9.87 × 10^4^ to 9.87 × 10^5^ CFU/reaction for the spiked fish; 9.87 × 10^3^ CFU/reaction for the spiked shrimp; 9.87 × 10^5^ CFU/reaction for the spiked shellfish samples (Table 5, Figure 4).

Similarly, when *V. parahaemolyticus* 17802 was spiked into the samples, for the target gene *ureC*, the observed LODs using the PCR assay were 9.17 × 10^4^ to 9.17 × 10^5^ CFU/reaction for the spiked fish; 9.17 × 10^4^ CFU/reaction for the spiked shrimp; 9.17 × 10^2^ CFU/reaction for the spiked shellfish samples (Table 5, Figure 4).

These results demonstrated that the sensitivity of the PCR assay was 1.00 × 10^1^- to 1.00 × 10^5^-fold lower than that of the LAMP method for the detection of the *opaR*, *tlh*, *ureC*, and *vpadF* genes in the spiked aquatic product samples.

### 2.6. Reproducibility of the LAMP Method

For the target genes *opaR*, *tlh*, *ureC*, and *vpadF* of *V. parahaemolyticus*, all the positive results could be repeated in all the tests performed for the detection of cell culture and genomic DNA samples, as well as of the spiked aquatic product samples, indicating high reproductivity (100%) of the LAMP method developed in this study.

### 2.7. Detection of Drinking Water and Aquatic Product Samples by the LAMP Method

Water samples from various sources were collected in Shanghai, China, in August 2021, including mineral water (*n* = 3), tap water (*n* = 3), river water (*n* = 3), lake water (*n* = 3), and estuarine water (*n* = 3). The samples were promptly screened by the LAMP method for the virulence-associated genes *opaR*, *vpadF*, *tlh*, and *ureC* of *V. parahaemolyticus*. As shown in Table 6, all the water samples tested negative for the target genes.

In addition, six species of commonly consumed aquatic product samples were collected from the fish market in Shanghai, China, in September 2021, and were examined by the LAMP method. The results showed that all the meat samples and most the intestine samples were negative for the *opaR*, *vpadF*, *tlh*, and *ureC* genes of *V. parahaemolyticus*. However, the *opaR* and *tlh* genes were detected in the intestine sample of *P. pekinensis* (Table 6, Figure 5), which were confirmed by routine microbial isolation and identification methods [35] (data not shown). These results suggested the risk of potentially virulent *V. parahaemolyticus* strains in the fish product.

## 3. Discussion

*V. parahaemolyticus* is the most prevalent gastroenteritis-causing pathogen in Asian countries [36,37]. Appropriate tools for the diagnosis of *V. parahaemolyticus* contamination in drinking water and aquatic products are the key to fight against outbreaks of the disease [38]. In this study, for the first time, we successfully developed a LAMP method for the detection of toxins and infection process-associated genes *opaR*, *tlh*, *ureC*, and *vpadF* of *V. parahaemolyticus*. Our data demonstrated high specificity of the inner, outer, and loop primers designed for each of the target genes in this study. No cross-reaction was observed with the other 7 species of *Vibrios* and 20 species of non-*Vibrios* strains, including common pathogenic bacteria, such as *V. cholerae*, *V**. alginolyticus*, *V**. fluvialis*, *V**. harvey**i*, and *V**. vulnificus*, as well as *L. monocytogenes*, *K. pneumonia**e*, *K. oxytoca*, *A**. hydrophila,* and *S**. aureus*.

Different sensitivity of the LAMP technique has been reported in the detection of foodborne pathogens; for example, Anupama et al. reported LODs of 1 pg/reaction and 1 CFU/reaction when targeting the *tdh* and *trh* genes of *V. parahaemolyticus* by LAMP [3]. The *toxR*-LAMP assay was able to detect 47–470 *V. parahaemolyticus* cells per reaction tube [32]. The LODs of the LAMP assays targeting the *rpoD* and *toxR* genes of *V. parahaemolyticus* were 3.7 and 450 CFU per test, respectively [31]. For the artificially contaminated seafood and seawater, the LODs of the LAMP assay were 120 and 150 fg per reaction for the *groEL* gene of *V. parahaemolyticus* [33]. In this study, inclusivity tests were conducted for each of the target genes with 50 *V. parahaemolyticus* strains. In a 25 µL LAMP system, the lowest observed LODs were 14.6 fg/reaction and 0.0103 CFU/reaction when targeting the *opaR* gene; 1.85 × 10^−4^ ng/reaction and 1.73 CFU/reaction when targeting the *vpadF* gene; 1.85 × 10^−4^ ng/reaction and 1.37 CFU/reaction when targeting the *tlh* gene; 1.85 pg/reaction and 0.44 CFU/reaction when targeting the *ureC* gene. The LAMP method developed in this study was more sensitive, with lower LODs, than the previous reports [3,31,32,33], not only for the detection of genomic DNA, but also for bacterial cell samples in water.

The influence of different aquatic product matrixes on the sensitivity of the LAMP method was observed in this study; for example, when the cell culture of *V. parahaemolyticus* ATCC17802 (2.75 × 10^9^–2.75 CFU/mL) was spiked into the aquatic product samples, the observed LODs ranged from 9.17 × 10^3^ to 9.17 × 10^1^ CFU/reaction when targeting the *ureC* gene for the spiked fish, shrimp, and shellfish samples, which was 2.08× 10^2^- to 2.08× 10^4^-fold lower than those obtained for the detection of *V. parahaemolyticus* cells in water (4.40 × 10^−1^ CFU/reaction). Moreover, our data revealed that the *L. vannamei* matrix appeared to interfere with the LAMP method more than those from the fish and shellfish. The *L. vannamei* matrix contained higher contents of proteins (23.3%) and crude fat (15.09%) [39] than the *P. pekinensis* (15.6% and 6.6%, respectively) and *M. edulis* (10.8% and 1.4%, respectively) matrixes [40], which may explain the observation. It will be interesting to investigate possible components in the *L. vannamei* matrix that contributed to the influence.

A comparison of the sensitivity of the LAMP method with the standard PCR assay revealed that the lowest LODs of the PCR assay ranged from 1.96 × 10^−2^ to 3.79 × 10^2^ ng/reaction and 1.37 × 10^3^ to 7.13 × 10^6^ CFU/reaction when targeting the *opaR* gene of *V. parahaemolyticus*; 3.05 × 10^−2^ to 6.04 × 10^2^ ng/reaction and 7.90 × 10^1^ to 9.20 × 10^6^ CFU/reaction when targeting the *tlh* gene; 1.85 × 10^−1^ ng/reaction and 4.40 × 10^0^ CFU/reaction when targeting the *ureC* gene; 2.68 × 10^−1^ to 4.16 × 10^2^ ng/reaction and 1.73 × 10^2^ to 8.63 × 10^5^ CFU/reaction when targeting the *vpadF* gene. These LODs were 1.00 × 10^1^- to 1.00 × 10^6^-fold, for the genomic DNA, and 1.00 × 10^1^- to 1.00 × 10^7^-fold, for the cell culture, lower than those obtained using the LAMP method. Although the aquatic product matrixes interfered with the sensitivity of the LAMP method, it was still more sensitive than the PCR assay.

The main limitation of the LAMP-based method is the complexity of the primer design to achieve the specificity of the detection. Another possible limitation of this method is that it could generate false-positive results due to the carry-over from previous experiments (due to its high sensitivity), especially when upgraded to an automated platform [41]. However, compared with the routine PCR and RT-PCR assays, the LAMP method developed in this study can be performed with a simple dry or water bath, which is more suitable for laboratories with less equipment [28]. Moreover, unlike the former two assays, the LAMP method does not require the reaction tubes to be opened, so there is no probable cross-contamination, and this method supports the field screening of potentially virulent *V. parahaemolyticus* with a larger diagnostic capacity.

## 4. Materials and Methods

### 4.1. Bacterial Strains and Culture Conditions

Bacterial strains used in this study are listed in Appendix A. Culture media were purchased as described previously [28]. *Vibrio* strains were incubated in 3% NaCl, pH 8.5 media, while non-*Vibrio* strains were incubated in 1% NaCl, pH 7.0 media [28].

### 4.2. Genomic DNA Preparation

Bacterial genomic DNA was prepared using TIANamp Bacterial Genomic DNA Extraction kit DP302 (Tiangen Biotech (Beijing) Co., Ltd., Beijing, China), or extracted by a thermal lysis method [28] with minor modifications. Briefly, 100 µL of bacteria cell culture was added into 900 µL 1 x phosphate-buffered saline (PBS, pH 7.4–7.6; Shanghai Sangon Biological Engineering Technology and Services Co., Ltd., Shanghai, China), mixed well, and then serially diluted. Cell pellet of each dilution was collected by centrifugation, resuspended with 200 µL sterile ultrapure water. The cell suspension was heated at 95 °C for 10 min, and then transferred onto ice for cooling. After centrifugation at 12,000 rpm for 5 min, the resulting lysis solution was used as DNA template. Extracted DNA samples were analyzed, and DNA concentrations and purity (A260/A280) were determined as described previously [28].

### 4.3. Designing of LAMP Primers

Sequences of target genes (*opaR*, *tlh*, *ureC*, and *vpadF*) in *V.*
*parahaemolyticus* were retrieved from National Center for Biotechnology Information (NCBI, https://www.ncbi.nlm.nih.gov/; accessed on 9 December 2020 to 23 May 2021) with GenBank accession numbers listed in Appendix A. The FIP and BIP, F3 and B3, and LF and LB primers targeting conserved sequences of each gene were designed using Primer Explorer Version 5 and SnapGene Viewer version 4.1.4 software (GSL Biotech LLC, Chicago, IL, USA) as described previously [28]. All primers (Table 1) were synthesized and DNA sequencing of PCR products was carried out by Sangon (Shanghai, China).

Following this, 1.6 µM of FIP and BIP primers, 0.05 µM of F3 and B3 primers, and 0.20 µM of LF and LB primers were used in a 25 µL LAMP reaction system. The system also contained 6 mM Mg^2+^, 1.0 mM dNTP, 8 units of *bacillus stearothermophilus (Bst)* DNA polymerase, and MnCl_2_ (15.60 mM)–calcein (1.30 mM) [27,28]. The one-step LAMP reaction was performed at 65 °C for 45 min.

### 4.4. Determination of Specificity and Sensitivity of the LAMP Method

Exclusivity (determined as 100% negative detection of non-target strains) and inclusivity (determined as 100% positive detection of target strains) tests of the LAMP method were determined as described previously [27,28]. The 50 bacterial strains and 51 *V. parah**aemolyticus* strains used in this study for the exclusivity and inclusivity tests are listed in Appendix A and Table 2, Table 3, Table 4, respectively. Genomic DNA samples extracted from each of these strains were serially diluted with DNase-/RNase-free deionized water (Tiangen Biotech Co., Ltd., Beijing, China), and used as DNA templates.

For the detection of *V. parah**a**emolyticus* cells, overnight cultures of each *V. parah**aemolyticus* strain were inoculated (1%, *v/v*) into fresh media (Appendix A) and bacterial cells grown at mid-logarithmic phase were harvested by centrifugation, resuspended, diluted, and enumerated as described previously [27,28].

### 4.5. Preparation and Analysis of Spiked Samples by LAMP

Spiked aquatic product samples were prepared according to the method described previously [42]. Fresh fish (*A. nobilis*, *C. auratus*, *C. idellus*, and *P. pekinensis*) (*n* = 3 per fish species, >500 g/sample), shrimp *(L. vannamei*, 500 g), and shellfish (*M. edulis*, 500 g) were purchased from Huangweixing aquatic product market in Nanhui New Town, Pudong New Area, Shanghai, China in September 2021. Twenty-five grams (wet weight) of mussel samples without skin or shell, or intestine samples, were cut with a sterile scalpel and homogenized with 225 mL of 1 × PBS (pH 7.4–7.6, Sangon, China) using BagMixer 400 (Interscience, Paris, France) homogenizer. Only the homogenate samples that were detected as negative for *V.*
*parah**aemolyticus* and the virulence-associated genes were used in the following spiked experiments [28,35].

Serial 10-fold dilutions of *V.*
*parah**aemolyticus* ATCC17802 and N7-19 culture were prepared, and calculated by plate counting method [28]. Further, 100 µL of each dilution was spiked into 900 µL fresh homogenate. Two microliters of 10-fold dilution of the mixture was used for the LAMP method [28].

### 4.6. PCR Assay

Primers used for the PCR assay in this study are listed in Table 1. The 10 µL PCR reaction solutions were prepared, and 30 cycles of PCR reactions were performed using Mastercycler Rpro PCR thermal cycler (Eppendorf, Hamburg, Germany), according to the methods described previously [28]. Amplicons were analyzed by agarose gel electrophoresis, then visualized and recorded [28].

### 4.7. Sample Collection and Analysis

Water samples were collected from various sources in August 2021 in Shanghai, China (Table 5). Freshwater fish (*A. nobilis*, *C. auratus*, *C. idellus*, and *P. pekinensis*) (*n* = 3 per fish species, >500 g/sample), shrimp (*L. vannamei*, 500 g), and shellfish (*M. edulis*, 500 g) samples were collected from the local aquatic product market as described above. All samples were maintained at 4 °C, immediately transported to the laboratory in Shanghai Ocean University (Shanghai, China), and analyzed according to the National Standards of the People’s Republic of China; we used the standard inspection methods for drinking water, the collection and preservation of water samples (GB/T 5750.2-2006), the direct processing of samples (SN/T 2332-2009), and the methods described previously by our research group [27,28].

## 5. Conclusions

*Vibrio parah**aemolyticus* can cause acute gastroenteritis, wound infection, and septicemia in humans. The bacterium is found growing in aquatic environments worldwide. The detection of *V. parahaemolyticus* in drinking water and aquatic products is essential for food safety control and human health. In this study, a simple, specific, and user-friendly diagnostic tool was developed for the first time, for the qualitative and quantitative detection of toxins and infection process-associated genes *opaR*, *vpadF*, *tlh,* and *ureC* in *V. parahaemolyticus,* using the LAMP technique. Three pairs of specific inner, outer, and loop primers were designed for targeting each of these genes, and the results showed no cross-reaction with the other common *Vibrios* and non-*Vibrios* pathogenic bacteria. Positive results in the one-step LAMP reaction (at 65 °C for 45 min) were identified by a change to light green and the emission of bright green fluorescence under visible light and UV light (302 nm), respectively. The lowest limit of detection (LOD) of the LAMP method for the target genes ranged from 1.46 × 10^−5^ to 1.85 × 10^−3^ ng/reaction (25 µL) for the genomic DNA, and from 1.03 × 10^−2^ to 1.73 × 10^0^ CFU/reaction (25 µL) for the cell culture of *V. parahaemolyticus*, which were 1.00 × 10^1^–1.00 × 10^6^ and 1.00 × 10^1^–1.00 × 10^7^ more sensitive than the standard polymerase chain reaction (PCR) assay. Similarly high efficiency was observed for the detection of spiked aquatic product samples. Water from various sources, and commonly consumed fish (*A. nobilis*, *C. auratus*, *C. idellus*, and *P. pekinensis*), shrimp (*L. vannamei*), and shellfish (*M. edulis*) samples, were promptly screened by the LAMP method, and *V. parahaemolyticus* was detected by the presence of *opaR* and *tlh* genes in the intestine of *P**. pekinensis*, indicating a risk of potentially pathogenic *V. parahaemolyticus* in the fish product. Overall, this study provides a molecular diagnostic tool for early diagnosis, particularly for the large-scale screening of drinking water and aquatic products contaminated by *V. parahaemolyticus*, a leading sea foodborne pathogen worldwide.

## Figures and Tables

**Figure 1 pathogens-11-00010-f001:**
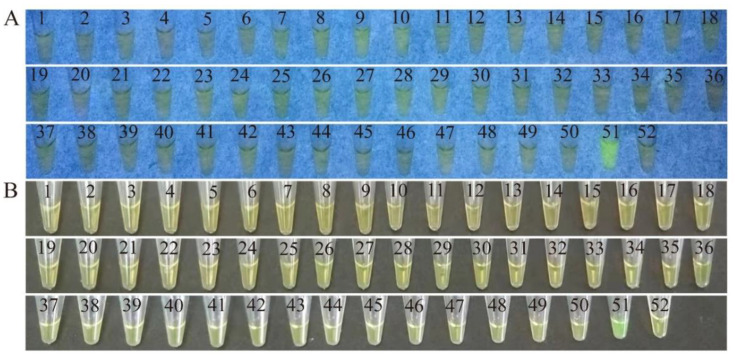
Specificity of the LAMP method targeting the *opaR* gene in the 50 bacterial strains (**A** to **B**). The results observed by the naked eye under visible light (**A**), and under UV light (302 nm) (**B**). Tubes 1 to 50: *Vibrio alginolyticus* ATCC17749, *V. alginolyticus* ATCC33787, *Vibrio fluvialis* ATCC33809, *Vibrio harveyi* ATCC BAA-1117, *V. harvey**i* ATCC33842, *Vibrio metschnikovii* ATCC700040, *Vibrio mimicus* bio-56759, *Vibrio vulnificus* ATCC27562, *V. vulnificus, Aeromonas hydrophila* ATCC35654, *A. hydrophila, Enterobacter cloacae* ATCC13047, *E. cloacae, Escherichia coli* ATCC8739, *E. coli* ATCC25922, *E. coli* K12, *Enterobacter sakazakii* CMCC45401, *Klebsiella oxytoca* 0707-27, *Klebsiella pneumoniae* 0717-1, *K. pneumoniae* 1202, *Klebsiella variicola* 0710-01, *Lactobacillus casei* D31, *Lactobacillus casei* T9, *L. casei* K17, *Listeria monocytogenes* ATCC19115, *Pseudomonas aeruginosa* ATCC9027, *P. aeruginosa* ATCC27853, *Salmonella enterica subsp. Enterica-Leminor et popoff* ATCC13312, *Staphylococcus aureus* ATCC25923, *S. aureus* ATCC 8095, *S. aureus* ATCC29213, *S. aureus* ATCC6538, *S. aureus* ATCC6538P, *Shigella dysenteriae* CMCC51252, *Salmonella* spp., *Shigella flexneri* CMCC51572, *S. flexneri* ATCC12022, *S. flexneri* CMCC51574, *Salmonella paratyphi-A*CMCC50093, *Shigella sonnei* ATCC25931, *Shigella sonnet* CMCC51592, *Salmonella typhimurium* ATCC15611, *Staphylococcus aureus*, *Vibrio cholerae* TCC39315 (N16961), *V. cholerae* GIM1.449, *V. cholerae* 805-38, *V. cholerae* 717-01, *V. cholerae* 805-29, *V. cholerae* 805-32, *V. cholerae* 717-25, respectively; tube 51: positive control *V. parahaemolyticus* ATCC17802 (*opaR^+^/vpadF^+^/tlh*^+^*/**ureC^+^*); tube 52: negative control.

**Figure 2 pathogens-11-00010-f002:**
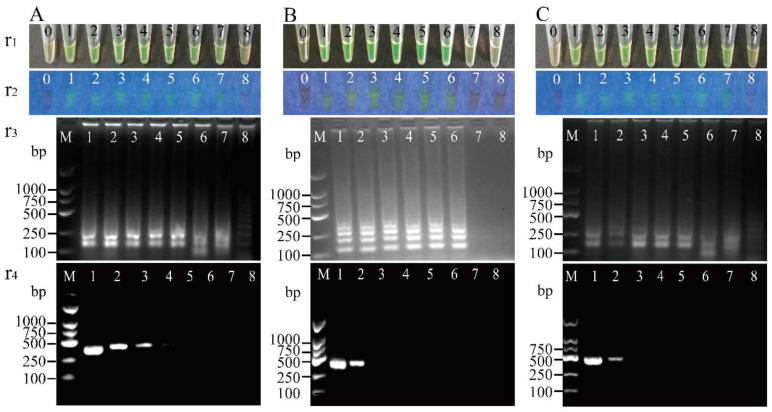
Sensitivity of the LAMP method targeting the *opaR* gene of *V. parahaemolyticus* cell culture. The results from the LAMP method were observed by the naked eye under visible light (r1) and UV light (302 nm) (r2), and verified by 2% agarose gel electrophoresis analysis (r3). r4: the results from the PCR assay. Lane M: DNA molecular weight marker (D2000 bp, Sangon, China). Tubes/lanes 1 to 8: contained serial dilutions of *V. parahaemolyticus* B4-13 cells (**A**); *V. parahaemolyticus* B11-3 cells (**B**); *V. parahaemolyticus* N3-3 cells (**C**). Tube 0: negative control.

**Figure 3 pathogens-11-00010-f003:**
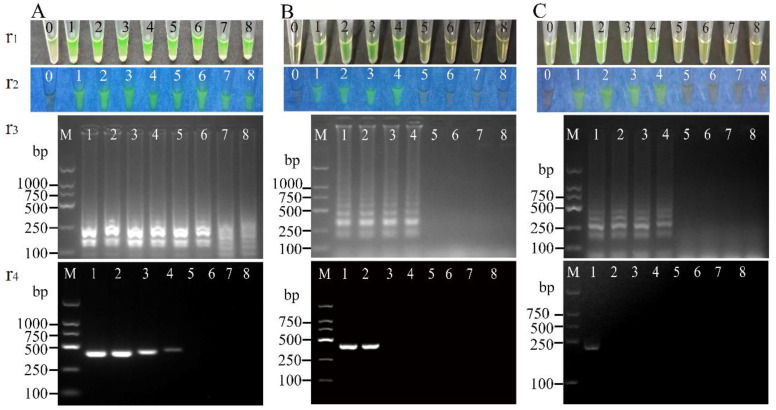
Sensitivity of the LAMP method targeting the *opaR*, *vpadF*, and *tlh* genes of *V. parahaemolyticus* genomic DNA. The results from the LAMP method were observed by the naked eye under visible light (r1) and UV light (302 nm) (r2), and verified by 2% agarose gel electrophoresis analysis (r3). r4: the results from the PCR assay. Lane M: DNA molecular weight marker (D2000 bp). Tubes/lanes 1 to 8: contained serial dilutions of genomic DNA samples extracted from *V. parahaemolyticus* L5-1 (**A**); *V. parahaemolyticus* N5-15 (**B**); *V. parahaemolyticus* N10-48 (**C**). Tube 0: negative control.

**Figure 4 pathogens-11-00010-f004:**
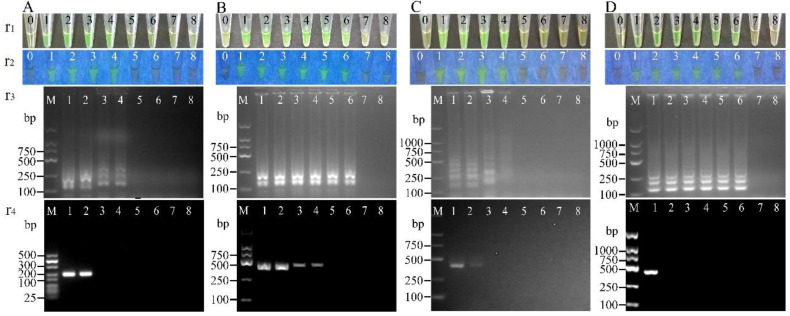
Sensitivity of the LAMP method for the detection of virulence-associated genes in spiked aquatic product samples. (**A**–**D**): the detection of *opaR* gene in *C. idella*, *vpadF* gene in *P. pekinensis*, *tlh* gene in *L. vannamei*, and *ureC* gene in *M. edulis* samples spiked with *V. parahaemolyticus* N7-19 (2.96 × 10^8^–2.96 CFU/mL) (**A**–**C**), and *V. parahaemolyticus* ATCC17802 (2.75 × 10^8^–2.75 CFU/mL). The results were observed by the naked eye under the visible light (r1) and the UV light (302 nm) (r2), and verified by 2% agarose gel electrophoresis analysis (r3) by the LAMP method. r4: the results by the PCR assay. Lane M: DNA molecular weight Marker (D2000 bp, and 25-500 bp, Sangon, China). Tube 0: negative control.

**Figure 5 pathogens-11-00010-f005:**
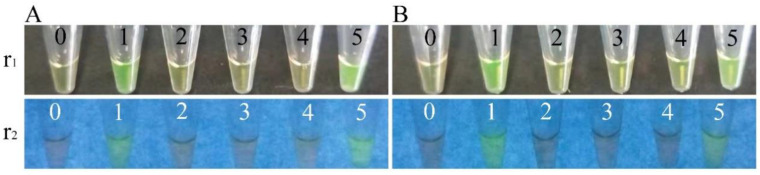
The detection of *opaR* (**A**) and *tlh* (**B**) genes in aquatic product samples by the LAMP method. The results were observed by the naked eye under visible light (r1) and UV light (302 nm) (r2). Tubes 0 to 5: negative control, positive control, and intestine samples of *A. nobilis*, *C. auratus*, *C. idella*, and *P. pekinensis*, respectively.

**Table 1 pathogens-11-00010-t001:** The primers designed and used in this study.

Primer	Target Gene	Reaction	Sequence (5′–3′)	Product Size (bp)	Source
FIP-*opaR*	*opaR*	LAMP	CAGTGACAATCTTGGCTTACGA-CGTGAAAACATCGCAAACA		This study
BIP-*opaR*	*opaR*	LAMP	GTTCGAGTGGAGCGCATCAA-TGGTTAGTGCGGTTGGTA		
F3-*opaR*	*opaR*	LAMP	TATCGACCTAGACATACACG		
B3-*opaR*	*opaR*	LAMP	TTCAATCGCTTTAATGAACATG		
LF-*opaR*	*opaR*	LAMP	CTCGATCATCGCATTGGTG		
LB-*opaR*	*opaR*	LAMP	TTCGAGTGGAGCGCATCAAC		
F-*opaR*	*opaR*	PCR	GTGGTGGTCACGCAGATA	417	This study
B-*opaR*	*opaR*	PCR	CGAACAGCGAGTAACAAA		
FIP-*vpadF*	*vpadF*	LAMP	CACGCCTGCGGTATTAGTGAGTACCACCAAAGGCTTATGTGT		This study
BIP-*vpadF*	*vpadF*	LAMP	ACCATGCGTACTGGTTAAGCCAACCGCACAAGATGAGGGT		
F3-*vpadF*	*vpadF*	LAMP	TCGCTCAACGTTCCCATG		
B3-*vpadF*	*vpadF*	LAMP	TTGTAGCGTTGTCATGCCA		
LF-*vpadF*	*vpadF*	LAMP	ACCGCACTGGAAATGCC		
LB-*vpadF*	*vpadF*	LAMP	TCAAGCTCGGCATAGAT		
F-*vpadF*	*vpadF*	PCR	TGCGGTATTAGTGAGTATGG	198	This study
B-*vpadF*	*vpadF*	PCR	AACGCTGTTCCTTTATGTTT		
FIP-*tlh*	*tlh*	LAMP	CGCAATGCGTGGGTGTACATGTGGTTTCGTGAACGCGAGT		This study
BIP-*tlh*	*tlh*	LAMP	CTCTGAGTGTGCGGCGTCTGTGAGTTGCTGTTGTTGGGT		
F3-*tlh*	*tlh*	LAMP	CTTCTGCGCCAGAAGAGC		
B3-*tlh*	*tlh*	LAMP	TTTCTCTGCGACATAGCGG		
LF-*tlh*	*tlh*	LAMP	CGGTTGATGTCCAAACAAG		
LB-*tlh*	*tlh*	LAMP	AAGTTTGTGTTCTGGGATG		
F-*tlh*	*tlh*	PCR	AGAACTTCATCTTGATGACACTGC	401	[34]
B-*tlh*	*tlh*	PCR	GCTACTTTCTAGCATTTTCTCTGC		
FIP-*ureC*	*ureC*	LAMP	GCCAGGGGTGACTGTTGTAGCTTTTATCGGTGGTGGCACTG		This study
BIP-*ureC*	*ureC*	LAMP	TGTTGGAAGCAGTCGATGAGCTCGCTTCTGGTTGACTCACA		
F3-*ureC*	*ureC*	LAMP	GGCTTGTCATCGGGTGTC		
B3-*ureC*	*ureC*	LAMP	GCTTCAATCTGCTCACGGAT		
LF-*ureC*	*ureC*	LAMP	ATTAGTACCAGCTACAGGG		
LB-*ureC*	*ureC*	LAMP	ATCAACGTCGGGCTATTC		
F-*ureC*	*ureC*	PCR	GACAAAGCCAAGTGACGA	312	This study
B-*ureC*	*ureC*	PCR	CAGTGCCACCACCGATAA		

**Table 2 pathogens-11-00010-t002:** Sensitivity of the LAMP method targeting the *opaR* gene in genomic DNA and cell culture of *V. parahaemolyticus* strains and comparison with the PCR assay.

Strain	Target Gene	Genomic DNA Dilutions (ng/μL)	LOD (ng/Reaction)		Rate of LOD for Genomic DNA (LAMP/PCR)	Cell Culture Dilutions (CFU/mL)	LOD (CFU/Reaction)		Rate of LOD for Cell Culture (LAMP/PCR)
			LAMP	PCR			LAMP	PCR	
B1-22	*opaR*	4.95 × 10^2^–4.95 × 10^−5^	9.90 × 10^−3^	9.90 × 10^−2^	1.00 × 10^1^	1.56 × 10^9^–1.56	5.20 × 10^3^	5.20 × 10^4^	1.00 × 10^1^
B3-8	*opaR*	1.08 × 10^2^–1.08 × 10^−5^	2.16 × 10^−1^	2.16 × 10^1^	1.00 × 10^2^	1.20 × 10^9^–1.20	4.00 × 10^2^	4.00 × 10^6^	1.00 × 10^4^
B4-13	*opaR*	1.83 × 10^2^–1.83 × 10^−5^	3.66 × 10^−2^	3.66 × 10^−1^	1.00 × 10^1^	1.32 ×10^9^–1.32	4.40 × 10^0^	4.40 × 10^4^	1.00 × 10^4^
B4-28	*opaR*	1.34 × 10^2^–1.34 × 10^−5^	2.68 × 10^−2^	2.68 × 10^1^	1.00 × 10^3^	1.34 × 10^8^–1.34	4.47 × 10^1^	4.47 × 10^5^	1.00 × 10^4^
B6-13	*opaR*	3.02 × 10^2^–3.02 × 10^−5^	6.04 × 10^−1^	6.04 × 10^0^	1.00 × 10^1^	1.25 × 10^9^–1.25	4.17 × 10^3^	4.20 × 10^4^	1.00 × 10^1^
B7-16	*opaR*	2.59 × 10^2^–2.59 × 10^−5^	5.18 × 10^−4^	5.18 × 10^0^	1.00 × 10^4^	2.34 × 10^8^–2.34	7.80 × 10^2^	7.80 × 10^4^	1.00 × 10^2^
B9-31	*opaR*	6.58 × 10^1^–6.58 × 10^−6^	1.32 × 10^−2^	1.32 × 10^1^	1.00 × 10^3^	1.16 × 10^9^–1.16	3.87 × 10^2^	3.87 × 10^4^	1.00 × 10^2^
B9-42	*opaR*	1.60 × 10^2^–1.60 × 10^−5^	3.21 × 10^−4^	3.21 × 10^0^	1.00 × 10^4^	5.20 × 10^7^–5.20	1.73 × 10^2^	1.73 × 10^5^	1.00 × 10^3^
B10-61	*opaR*	2.30 × 10^2^–2.30 × 10^−5^	4.61 × 10^−3^	4.61 × 10^−1^	1.00 × 10^2^	6.40 × 10^8^–6.40	2.13 × 10^2^	2.13 × 10^4^	1.00 × 10^2^
B11-3	*opaR*	1.41 × 10^2^–1.41 × 10^−5^	2.83 × 10^−2^	2.83 × 10^0^	1.00 × 10^2^	7.00 × 10^8^–7.00	2.33 × 10^1^	2.33 × 10^5^	1.00 × 10^4^
L5-1	*opaR*	2.10 × 10^2^–2.10 × 10^−5^	4.21 × 10^−5^	4.21 × 10^−1^	1.00 × 10^4^	1.71 × 10^8^–1.71	5.70 × 10^1^	5.70 × 10^4^	1.00 × 10^3^
L7-7	*opaR*	2.15 × 10^2^–2.15 × 10^−5^	4.30 × 10^−1^	4.30 × 10^1^	1.00 × 10^2^	2.14 × 10^9^–2.14	7.13 × 10^3^	7.13 × 10^6^	1.00 × 10^3^
L7-45	*opaR*	1.24 × 10^2^–1.24 × 10^−5^	2.49 × 10^−4^	2.49 × 10^0^	1.00 × 10^4^	1.29 × 10^9^–1.29	4.30 × 10^3^	4.30 × 10^5^	1.00 × 10^2^
L10-15	*opaR*	1.49 × 10^2^–1.49 × 10^−5^	2.98 × 10^−3^	2.98 × 10^−1^	1.00 × 10^2^	2.59 × 10^8^–2.59	8.63 × 10^1^	8.63 × 10^4^	1.00 × 10^3^
N2-8	*opaR*	2.36 × 10^2^–2.36 × 10^−5^	4.72 × 10^−3^	4.72 × 10^−1^	1.00 × 10^2^	3.10 × 10^8^–3.10	1.03 × 10^3^	1.03 × 10^5^	1.00 × 10^2^
N2-11	*opaR*	9.36 × 10^1^–9.36 × 10^−6^	1.87 × 10^−1^	1.87 × 10^1^	1.00 × 10^2^	7.40 × 10^7^–7.40	2.47 × 10^3^	2.47 × 10^4^	1.00 × 10^1^
N2-20	*opaR*	1.43 × 10^2^–1.43 × 10^−5^	2.85 × 10^−3^	2.85 × 10^−1^	1.00 × 10^2^	1.14 × 10^9^–1.14	3.80 × 10^2^	3.80 × 10^3^	1.00 × 10^1^
N2-25	*opaR*	1.44 × 10^2^–1.44 × 10^−5^	2.89 × 10^−5^	2.89 × 10^−1^	1.00 × 10^4^	6.40 × 10^8^–6.40	2.13 × 10^3^	2.13 × 10^5^	1.00 × 10^2^
N3-2	*opaR*	1.20 × 10^2^–1.20 × 10^−5^	2.39 × 10^−5^	2.39 × 10^1^	1.00 × 10^6^	1.23 × 10^9^–1.23	4.10 × 10^2^	4.10 × 10^4^	1.00 × 10^2^
N3-3	*opaR*	1.11 × 10^2^–1.11 × 10^−5^	2.22 × 10^−5^	2.22 × 10^0^	1.00 × 10^5^	9.10 × 10^7^–9.10	3.03 × 10^−1^	3.03 × 10^4^	1.00 × 10^5^
N3-11	*opaR*	1.67 × 10^2^–1.67 × 10^−5^	3.35 × 10^−2^	3.35 × 10^−1^	1.00 × 10^1^	8.00 × 10^8^–8.00	2.67 × 10^1^	2.67 × 10^5^	1.00 × 10^4^
N3-13	*opaR*	1.95 × 10^2^–1.95 × 10^−5^	3.90 × 10^−4^	3.90 × 10^−2^	1.00 × 10^2^	6.60 × 10^7^–6.60	2.20 × 10^1^	2.20 × 10^4^	1.00 × 10^3^
N3-29	*opaR*	1.29 × 10^2^–1.29 × 10^−5^	2.57 × 10^−5^	2.57 × 10^−1^	1.00 × 10^4^	7.50 × 10^7^–7.50	2.50 × 10^0^	2.50 × 10^5^	1.00 × 10^5^
N3-30	*opaR*	1.47 × 10^2^–1.47 × 10^−5^	2.94 × 10^−1^	2.94 × 10^1^	1.00 × 10^2^	2.10 × 10^8^–2.10	7.00 × 10^2^	7.00 × 10^4^	1.00 × 10^2^
N3-32	*opaR*	6.95 × 10^1^–6.95 × 10^−6^	1.39 × 10^−3^	1.39 × 10^−1^	1.00 × 10^2^	2.40 × 10^8^–2.40	8.00 × 10^1^	8.00 × 10^4^	1.00 × 10^3^
N3-33	*opaR*	9.60 × 10^1^–9.60 × 10^−6^	1.92 × 10^−2^	1.92 × 10^−1^	1.00 × 10^1^	3.80 × 10^7^–3.80	1.27 × 10^1^	1.27 × 10^5^	1.00 × 10^4^
N4-9	*opaR*	9.72 × 10^1^–9.72 × 10^−6^	1.94 × 10^−5^	1.94 × 10^−1^	1.00 × 10^4^	9.40 × 10^7^–9.40	3.13 × 10^0^	3.13 × 10^3^	1.00 × 10^3^
N4-26	*opaR*	7.31 × 10^1^–7.31 × 10^−6^	1.46 × 10^−5^	1.46 × 10^−1^	1.00 × 10^4^	8.30 × 10^7^–8.30	2.77 × 10^−1^	2.77 × 10^5^	1.00 × 10^6^
N4-31	*opaR*	9.37 × 10^1^–9.37 × 10^−6^	1.87 × 10^−5^	1.87 × 10^−1^	1.00 × 10^4^	2.70 × 10^8^–2.70	9.00 × 10^0^	9.00 × 10^3^	1.00 × 10^3^
N4-46	*opaR*	8.34 × 10^1^–8.34 × 10^−6^	1.67 × 10^−1^	1.67 × 10^1^	1.00 × 10^2^	8.60 × 10^7^–8.60	2.87 × 10^2^	2.87 × 10^5^	1.00 × 10^3^
N5-15	*opaR*	7.26 × 10^1^–7.26 × 10^−6^	1.45 × 10^−2^	1.45 × 10^−1^	1.00 × 10^1^	1.30 × 10^8^–1.30	4.33 × 10^0^	4.33 × 10^3^	1.00 × 10^3^
N6-7	*opaR*	1.79 × 10^2^–1.79 × 10^−5^	3.58 × 10^−5^	3.58 × 10^−2^	1.00 × 10^3^	1.61 × 10^8^–1.61	5.37 × 10^−2^	5.37 × 10^5^	1.00 × 10^7^
N6-10	*opaR*	1.21 × 10^2^–1.21 × 10^−5^	2.42 × 10^−2^	2.42 × 10^−1^	1.00 × 10^1^	1.41 × 10^8^–1.41	4.70 × 10^1^	4.70 × 10^4^	1.00 × 10^3^
N6-16	*opaR*	9.80 × 10^1^–9.80 × 10^−6^	1.96 × 10^−3^	1.96 × 10^−2^	1.00 × 10^1^	2.54 × 10^8^–2.4	8.47 × 10^1^	8.47 × 10^4^	1.00 × 10^3^
N6-26	*opaR*	1.20 × 10^2^–1.20 × 10^−5^	2.39 × 10^−2^	2.39 × 10^0^	1.00 × 10^2^	1.76 × 10^8^–1.76	5.87 × 10^1^	5.87 × 10^4^	1.00 × 10^3^
N7-3	*opaR*	8.79 × 10^1^–8.79 × 10^−6^	1.76 × 10^−2^	1.76 × 10^1^	1.00 × 10^3^	9.50 × 10^7^–9.50	3.17 × 10^0^	3.17 × 10^5^	1.00 × 10^5^
N7-9	*opaR*	1.53 × 10^2^–1.53 × 10^−5^	3.05 × 10^−3^	3.05 × 10^−1^	1.00 × 10^2^	3.60 × 10^8^–3.60	1.20 × 10^1^	1.20 × 10^5^	1.00 × 10^4^
N7-45	*opaR*	1.22 × 10^2^–1.22 × 10^−5^	2.43 × 10^−3^	2.43 × 10^0^	1.00 × 10^3^	2.76 × 10^8^–2.76	9.20 × 10^−2^	9.20 × 10^4^	1.00 × 10^6^
N7-19	*opaR*	1.54 × 10^2^–1.54 × 10^−5^	3.07 × 10^−5^	3.07 × 10^−1^	1.00 × 10^4^	1.02 × 10^9^–1.02	3.40 × 10^3^	3.40 × 10^4^	1.00 × 10^1^
N8-9	*opaR*	1.09 × 10^2^–1.09 × 10^−5^	2.18 × 10^−3^	2.18 × 10^0^	1.00 × 10^3^	1.34 × 10^8^–1.34	4.47 × 10^1^	4.47 × 10^5^	1.00 × 10^4^
N8-13	*opaR*	1.82 × 10^2^–1.82 × 10^−5^	3.64 × 10^−2^	3.64 × 10^−1^	1.00 × 10^1^	2.45 × 10^8^–2.45	8.17 × 10^−1^	8.17 × 10^4^	1.00 × 10^5^
N8-36	*opaR*	9.17 × 10^1^–9.17 × 10^−6^	1.83 × 10^−5^	1.83 × 10^−1^	1.00 × 10^4^	2.18 × 10^8^–2.18	7.27 × 10^−1^	7.27 × 10^5^	1.00 × 10^6^
N9-24	*opaR*	1.02 × 10^2^–1.02 × 10^−5^	2.05 × 10^−2^	2.05 × 10^−1^	1.00 × 10^1^	3.10 × 10^7^–3.10	1.03 × 10^−2^	1.03 × 10^5^	1.00 × 10^7^
N9-31	*opaR*	2.08 × 10^2^–2.08 × 10^−5^	4.16 × 10^−4^	4.16 × 10^0^	1.00 × 10^4^	2.60 × 10^8^–2.60	8.67 × 10^0^	8.67 × 10^3^	1.00 × 10^3^
N10-20	*opaR*	1.00 × 10^2^–1.00 × 10^−5^	2.00 × 10^−3^	2.00 × 10^−2^	1.00 × 10^1^	2.53 × 10^8^–2.53	8.43 × 10^0^	8.43 × 10^3^	1.00 × 10^3^
N10-48	*opaR*	1.16 × 10^2^–1.16 × 10^−5^	2.32 × 10^−3^	2.32 × 10^−2^	1.00 × 10^1^	2.56 × 10^8^–2.56	8.53 × 10^−2^	8.53 × 10^4^	1.00 × 10^6^
Q5-6	*opaR*	2.23 × 10^2^–2.23 × 10^−5^	4.46 × 10^−5^	4.46 × 10^0^	1.00 × 10^5^	1.68 × 10^8^–1.68	5.60 × 10^0^	5.60 × 10^5^	1.00 × 10^5^
Q8-2	*opaR*	8.83 × 10^1^–8.83 × 10^−6^	1.77 × 10^−2^	1.77 × 10^2^	1.00 × 10^4^	8.90 × 10^7^–8.90	2.97 × 10^1^	2.97 × 10^4^	1.00 × 10^3^
Q8-7	*opaR*	1.89 × 10^2^–1.90 × 10^−5^	3.79 × 10^−1^	3.79 × 10^2^	1.00 × 10^3^	4.10 × 10^7^–4.10	1.37 × 10^0^	1.37 × 10^3^	1.00 × 10^3^
ATCC 17802	*opaR*	9.26 × 10^1^–9.26 × 10^−6^	1.85 × 10^0^	1.85 × 10^2^	1.00 × 10^2^	1.32 × 10^8^–1.32	4.40 × 10^3^	4.40 × 10^5^	1.00 × 10^2^

**Table 3 pathogens-11-00010-t003:** Sensitivity of the LAMP method targeting the *vpadF* gene in genomic DNA and cell culture of *V. parahaemolyticus* strains and comparison with the PCR assay.

Strain	Target Gene	Genomic DNA Dilutions (ng/μL)	LOD (ng/Reaction)		Rate of LOD for Genomic DNA (LAMP/PCR)	Cell Culture Dilutions (CFU/mL)	LOD (CFU/Reaction)		Rate of LOD for Cell Culture (LAMP/PCR)
			LAMP	PCR			LAMP	PCR	
B1-22	*vpadF*	4.95 × 10^2^–4.95 ×10^−5^	9.90 × 10^−3^	9.90 × 10^0^	1.00 × 10^3^	1.56 × 10^9^–1.56	5.20 × 10^1^	5.20 × 10^4^	1.00 × 10^3^
B3-8	*vpadF*	1.08 × 10^2^–1.08 × 10^−5^	2.16 × 10^−3^	2.16 × 10^0^	1.00 × 10^3^	1.20 × 10^9^–1.20	4.00 × 10^2^	4.00 × 10^4^	1.00 × 10^2^
B4-13	*vpadF*	1.83 × 10^2^–1.83 × 10^−5^	3.66 × 10^−3^	3.66 × 10^0^	1.00 × 10^3^	1.32 × 10^9^–1.32	4.40 × 10^2^	4.40 × 10^4^	1.00 × 10^2^
B4-28	*vpadF*	1.34 × 10^2^–1.34 × 10^−5^	2.68 × 10^−4^	2.68 × 10^−1^	1.00 × 10^3^	1.34 × 10^8^–1.34	4.47 × 10^1^	4.47 × 10^3^	1.00 × 10^2^
B7-16	*vpadF*	2.59 × 10^2^–2.59 × 10^−5^	5.18 × 10^−2^	5.18 × 10^0^	1.00 × 10^2^	2.34 × 10^8^–2.34	7.80 × 10^1^	7.80 × 10^4^	1.00 × 10^3^
B9-31	*vpadF*	6.58 × 10^1^–6.58 × 10^−6^	1.32 × 10^−1^	1.32 × 10^1^	1.00 × 10^2^	1.16 × 10^9^–1.16	3.87 × 10^2^	3.87 × 10^4^	1.00 × 10^2^
B9-42	*vpadF*	1.60 × 10^2^–1.60 × 10^−5^	3.21 × 10^−1^	3.21 × 10^1^	1.00 × 10^2^	5.20 × 10^7^–5.20	1.73 × 10^0^	1.73 × 10^2^	1.00 × 10^2^
B11-3	*vpadF*	1.41 × 10^2^–1.41 × 10^−5^	2.83 × 10^−3^	2.83 × 10^0^	1.00 × 10^3^	7.00 × 10^8^–7.00	2.33 × 10^2^	2.33 × 10^4^	1.00 × 10^2^
L5-1	*vpadF*	2.10 × 10^2^–2.10 × 10^−5^	4.21 × 10^−1^	4.21 × 10^1^	1.00 × 10^2^	1.71 × 10^8^–1.71	5.70 × 10^2^	5.70 × 10^4^	1.00 × 10^2^
L7-7	*vpadF*	2.15 × 10^2^–2.15 × 10^−5^	4.30 × 10^−^^1^	4.30 × 10^1^	1.00 × 10^2^	2.14 × 10^9^–2.14	7.13 × 10^2^	7.13 × 10^4^	1.00 × 10^2^
L7-45	*vpadF*	1.24 × 10^2^–1.24 × 10^−5^	2.49 × 10^−3^	2.49 × 10^0^	1.00 × 10^3^	1.29 × 10^9^–1.29	4.30 × 10^3^	4.30 × 10^5^	1.00 × 10^2^
L10-15	*vpadF*	1.49 × 10^2^–1.49 × 10^−5^	2.98 × 10^−2^	2.98 × 10^1^	1.00 × 10^3^	2.59 × 10^8^–2.59	8.63 × 10^3^	8.63 × 10^5^	1.00 × 10^2^
N2-8	*vpadF*	2.36 × 10^2^–2.36 × 10^−5^	4.72 × 10^−3^	4.72 × 10^0^	1.00 × 10^3^	3.10 × 10^8^–3.10	1.03 × 10^2^	1.03 × 10^5^	1.00 × 10^3^
N2-11	*vpadF*	9.36 × 10^1^–9.36 × 10^−6^	1.87 × 10^−1^	1.87 × 10^1^	1.00 × 10^2^	7.40 × 10^7^–7.40	2.47 × 10^2^	2.47 × 10^4^	1.00 × 10^2^
N2-20	*vpadF*	1.43 × 10^2^–1.43 × 10^−5^	2.85 × 10^−3^	2.85 × 10^0^	1.00 × 10^3^	1.14 × 10^9^–1.14	3.80 × 10^2^	3.80 × 10^4^	1.00 × 10^2^
N3-2	*vpadF*	1.20 × 10^2^–1.20 × 10^−5^	2.39 × 10^−2^	2.39 × 10^0^	1.00 × 10^2^	1.23 × 10^9^–1.23	4.10 × 10^2^	4.10 × 10^4^	1.00 × 10^2^
N3-3	*vpadF*	1.11 × 10^2^–1.11 × 10^−5^	2.22 × 10^−2^	2.22 × 10^1^	1.00 × 10^3^	9.10 × 10^7^–9.10	3.03 × 10^2^	3.03 × 10^4^	1.00 × 10^2^
N3-11	*vpadF*	1.67 × 10^2^–1.67 × 10^−5^	3.35 × 10^−2^	3.35 × 10^1^	1.00 × 10^3^	8.00 × 10^8^–8.00	2.67 × 10^3^	2.67 × 10^5^	1.00 × 10^2^
N3-13	*vpadF*	1.95 × 10^2^–1.95 × 10^−5^	3.89 × 10^−2^	3.89 × 10^0^	1.00 × 10^2^	6.60 × 10^7^–6.60	2.20 × 10^1^	2.20 × 10^3^	1.00 × 10^2^
N3-29	*vpadF*	1.29 × 10^2^–1.29 × 10^−5^	2.57 × 10^−3^	2.57 × 10^0^	1.00 × 10^3^	7.50 × 10^7^–7.50	2.50 × 10^2^	2.50 × 10^4^	1.00 × 10^2^
N3-30	*vpadF*	1.47 × 10^2^–1.47 × 10^−5^	2.94 × 10^−1^	2.94 × 10^1^	1.00 × 10^2^	2.10 × 10^8^–2.10	7.00 × 10^2^	7.00 × 10^5^	1.00 × 10^3^
N3-32	*vpadF*	6.95 × 10^1^–6.95 × 10^−6^	1.39 × 10^−2^	1.39 × 10^0^	1.00 × 10^2^	2.40 × 10^8^–2.40	8.00 × 10^3^	8.00 × 10^5^	1.00 × 10^2^
N4-9	*vpadF*	9.72 × 10^1^–9.72 × 10^−6^	1.94 × 10^−3^	1.94 × 10^0^	1.00 × 10^3^	9.40 × 10^7^–9.40	3.13 × 10^3^	3.13 × 10^5^	1.00 × 10^2^
N4-26	*vpadF*	7.31 × 10^1^–7.31 × 10^−6^	1.46 × 10^−2^	1.46 × 10^1^	1.00 × 10^3^	8.30 × 10^7^–8.30	2.77 × 10^3^	2.77 × 10^4^	1.00 × 10^1^
N4-31	*vpadF*	9.37 × 10^1^–9.37 × 10^−6^	1.87 × 10^−1^	1.87 × 10^1^	1.00 × 10^2^	2.70 × 10^8^–2.70	9.00 × 10^2^	9.00 × 10^4^	1.00 × 10^2^
N4-46	*vpadF*	8.34 × 10^1^–8.34 × 10^−6^	1.67 × 10^−3^	1.67 × 10^0^	1.00 × 10^3^	8.60 × 10^7^–8.60	2.87 × 10^2^	2.87 × 10^4^	1.00 × 10^2^
N5-15	*vpadF*	7.26 × 10^1^–7.26 × 10^−6^	1.45 × 10^−2^	1.45 × 10^0^	1.00 × 10^2^	1.30 × 10^8^–1.30	4.33 × 10^3^	4.33 × 10^5^	1.00 × 10^2^
N6-16	*vpadF*	9.80 × 10^1^–9.80 × 10^−6^	1.96 × 10^−2^	1.96 × 10^1^	1.00 × 10^3^	2.54 × 10^8^–2.40	8.47 × 10^3^	8.47 × 10^4^	1.00 × 10^1^
N6-26	*vpadF*	1.20 × 10^2^–1.20 × 10^−5^	2.39 × 10^−3^	2.39 × 10^0^	1.00 × 10^3^	1.76 × 10^8^–1.76	5.87 × 10^2^	5.87 × 10^4^	1.00 × 10^2^
N7-45	*vpadF*	1.22 × 10^2^–1.22 × 10^−5^	2.43 × 10^−1^	2.43 × 10^1^	1.00 × 10^2^	2.76 × 10^8^–2.76	9.20 × 10^2^	9.20 × 10^4^	1.00 × 10^2^
N7-19	*vpadF*	1.54 × 10^2^–1.54 × 10^−5^	3.07 × 10^−4^	3.07 × 10^−1^	1.00 × 10^3^	1.02 × 10^7^–1.02	3.40 × 10^2^	3.40 × 10^5^	1.00 × 10^3^
N8-9	*vpadF*	1.09 × 10^2^–1.09 × 10^−5^	2.18 × 10^−3^	2.18 × 10^0^	1.00 × 10^3^	1.34 × 10^8^–1.34	4.47 × 10^2^	4.47 × 10^4^	1.00 × 10^2^
N8-13	*vpadF*	1.82 × 10^2^–1.82 × 10^−5^	3.64 × 10^−3^	3.64 × 10^0^	1.00 × 10^3^	2.45 × 10^8^–2.45	8.17 × 10^2^	8.17 × 10^4^	1.00 × 10^2^
N8-36	*vpadF*	9.17 × 10^1^–9.17 × 10^−6^	1.83 × 10^−1^	1.83 × 10^0^	1.00 × 10^1^	2.18 × 10^8^–2.18	7.27 × 10^2^	7.27 × 10^4^	1.00 × 10^2^
N9-24	*vpadF*	1.02 × 10^2^–1.02 × 10^−5^	2.05 × 10^−1^	2.05 × 10^1^	1.00 × 10^2^	3.10 × 10^7^–3.10	1.03 × 10^3^	1.03 × 10^4^	1.00 × 10^1^
N9-31	*vpadF*	2.08 × 10^2^–2.08 × 10^−5^	4.16 × 10^−1^	4.16 × 10^2^	1.00 × 10^3^	2.60 × 10^8^–2.60	8.67 × 10^1^	8.67 × 10^4^	1.00 × 10^3^
Q5-6	*vpadF*	2.23 × 10^2^–2.23 × 10^−5^	4.46 × 10^−4^	4.46 × 10^0^	1.00 × 10^4^	1.68 × 10^8^–1.68	5.60 × 10^2^	5.60 × 10^5^	1.00 × 10^3^
Q8-15	*vpadF*	1.10 × 10^2^–1.10 × 10^−5^	2.21 × 10^−2^	2.21 × 10^1^	1.00 × 10^3^	2.37 × 10^8^–2.37	7.90 × 10^2^	7.90 × 10^4^	1.00 × 10^2^
ATCC 17802	*vpadF*	9.26 × 10^1^–9.26 × 10^−6^	1.85 × 10^−^^4^	1.85 × 10^0^	1.00 × 10^4^	1.32 × 10^8^–1.32	4.40 × 10^2^	4.40 × 10^3^	1.00 × 10^1^

**Table 4 pathogens-11-00010-t004:** Sensitivity of the LAMP method targeting the *tlh* and *ureC* genes in genomic DNA and cell culture of *V. parahaemolyticus* strains and comparison with the PCR assay.

Strain	Target Gene	Genomic DNA Dilutions (ng/μL)	LOD (ng/Reaction)		Rate of LOD for Genomic DNA (LAMP/PCR)	Cell Culture Dilutions (CFU/mL)	LOD (CFU/Reaction)		Rate of LOD for Cell Culture (LAMP/PCR)
			LAMP	PCR			LAMP	PCR	
B1-22	*tlh*	4.95 × 10^2^–4.95 × 10^−5^	9.90 × 10^−2^	9.90 × 10^1^	1.00 × 10^3^	1.56 × 10^9^–1.56	5.20 × 10^1^	5.20 × 10^3^	1.00 × 10^2^
B3-8	*tlh*	1.08 × 10^2^–1.08 × 10^−5^	2.16 × 10^−1^	2.16 × 10^0^	1.00 × 10^1^	1.20 × 10^9^–1.20	4.00 × 10^2^	4.00 × 10^5^	1.00 × 10^3^
B4-13	*tlh*	1.83 × 10^2^–1.83 × 10^−5^	3.66 × 10^0^	3.66 × 10^1^	1.00 × 10^1^	1.32 × 10^9^–1.32	4.40 × 10^2^	4.40 × 10^5^	1.00 × 10^3^
B4-28	*tlh*	1.34 × 10^2^–1.34 × 10^−5^	2.68 × 10^1^	2.68 × 10^2^	1.00 × 10^1^	1.34 × 10^8^–1.34	4.47 × 10^1^	4.47 × 10^2^	1.00 × 10^1^
B6-13	*tlh*	3.02 × 10^2^–3.02 × 10^−5^	6.04 × 10^0^	6.04 × 10^2^	1.00 × 10^2^	1.25 × 10^9^–1.25	4.17 × 10^3^	4.17 × 10^5^	1.00 × 10^2^
B7-16	*tlh*	2.59 × 10^2^–2.59 × 10^−5^	5.18 × 10^−3^	5.18 × 10^1^	1.00 × 10^4^	2.34 × 10^8^–2.34	7.80 × 10^1^	7.80 × 10^2^	1.00 × 10^1^
B9-31	*tlh*	6.58 × 10^1^–6.58 × 10^−6^	1.32 × 10^−1^	1.32 × 10^0^	1.00 × 10^1^	1.16 × 10^9^–1.16	3.87 × 10^2^	3.87 × 10^5^	1.00 × 10^3^
B9-42	*tlh*	1.60 × 10^2^–1.60 × 10^−5^	3.21 × 10^0^	3.21 × 10^1^	1.00 × 10^1^	5.20 × 10^7^–5.20	1.73 × 10^3^	1.73 × 10^5^	1.00 × 10^2^
B10-61	*tlh*	2.30 × 10^2^–2.30 × 10^−5^	4.61 × 10^−3^	4.61 × 10^−1^	1.00 × 10^2^	6.40 × 10^8^–6.40	2.13 × 10^3^	2.13 × 10^6^	1.00 × 10^3^
B11-3	*tlh*	1.41 × 10^2^–1.41 × 10^−5^	2.83 × 10^−1^	2.83 × 10^2^	1.00 × 10^3^	7.00 × 10^8^–7.00	2.33 × 10^2^	2.33 × 10^5^	1.00 × 10^3^
L5-1	*tlh*	2.10 × 10^2^–2.10 × 10^−5^	4.21 × 10^−2^	4.21 × 10^0^	1.00 × 10^2^	1.71 × 10^8^–1.71	5.70 × 10^3^	5.70 × 10^5^	1.00 × 10^2^
L7-7	*tlh*	2.15 × 10^2^–2.15 × 10^−5^	4.29 × 10^−1^	4.29 × 10^1^	1.00 × 10^2^	2.14 × 10^9^–2.14	7.13 × 10^3^	7.13 × 10^6^	1.00 × 10^3^
L7-45	*tlh*	1.24 × 10^2^–1.24 × 10^−5^	2.49 × 10^0^	2.49 × 10^2^	1.00 × 10^2^	1.29 × 10^9^–1.29	4.30 × 10^2^	4.30 × 10^5^	1.00 × 10^3^
L10-15	*tlh*	1.49 × 10^2^–1.49 × 10^−5^	2.98 × 10^1^	2.98 × 10^2^	1.00 × 10^1^	2.59 × 10^8^–2.59	8.63 × 10^3^	8.63 × 10^5^	1.00 × 10^2^
N2-8	*tlh*	2.36 × 10^2^–2.36 × 10^−5^	4.72 × 10^−2^	4.72 × 10^−1^	1.00 × 10^1^	3.10 × 10^8^–3.10	1.03 × 10^2^	1.03 × 10^3^	1.00 × 10^1^
N2-11	*tlh*	9.36 × 10^1^–9.36 × 10^−6^	1.87 × 10^1^	1.87 × 10^2^	1.00 × 10^1^	7.40 × 10^7^–7.40	2.47 × 10^3^	2.47 × 10^5^	1.00 × 10^2^
N2-20	*tlh*	1.43 × 10^2^–1.43 × 10^−5^	2.85 × 10^−2^	2.85 × 10^1^	1.00 × 10^3^	1.14 × 10^9^–1.14	3.80 × 10^3^	3.80 × 10^6^	1.00 × 10^3^
N2-25	*tlh*	1.44 × 10^2^–1.45 × 10^−5^	2.89 × 10^0^	2.89 × 10^1^	1.00 × 10^1^	6.40 × 10^7^–6.40	2.13 × 10^3^	2.13 × 10^5^	1.00 × 10^2^
N3-2	*tlh*	1.20 × 10^2^–1.20 × 10^−5^	2.39 × 10^−2^	2.39 × 10^1^	1.00 × 10^3^	1.23 × 10^9^–1.23	4.10 × 10^2^	4.10 × 10^5^	1.00 × 10^3^
N3-3	*tlh*	1.11 × 10^2^–1.11 × 10^−5^	2.22 × 10^−1^	2.22 × 10^1^	1.00 × 10^2^	9.10 × 10^7^–9.10	3.03 × 10^3^	3.03 × 10^5^	1.00 × 10^2^
N3-11	*tlh*	1.67 × 10^2^–1.67 × 10^−5^	3.35 × 10^1^	3.35 × 10^2^	1.00 × 10^1^	8.00 × 10^8^–8.00	2.67 × 10^2^	2.67 × 10^5^	1.00 × 10^3^
N3-13	*tlh*	1.95 × 10^2^–1.95 × 10^−5^	3.90 × 10^−2^	3.90 × 10^1^	1.00 × 10^3^	6.60 × 10^7^–6.60	2.20 × 10^2^	2.20 × 10^5^	1.00 × 10^3^
N3-29	*tlh*	1.29 × 10^2^–1.29 × 10^−5^	2.57 × 10^1^	2.57 × 10^2^	1.00 × 10^1^	7.50 × 10^7^–7.50	2.50 × 10^3^	2.50 × 10^4^	1.00 × 10^1^
N3-30	*tlh*	1.47 × 10^2^–1.47 × 10^−5^	2.94 × 10^1^	2.94 × 10^2^	1.00 × 10^1^	2.10 × 10^8^–2.10	7.00 × 10^3^	7.00 × 10^4^	1.00 × 10^1^
N3-32	*tlh*	6.95 × 10^1^–6.95 × 10^−6^	1.39 × 10^−1^	1.39 × 10^1^	1.00 × 10^2^	2.40 × 10^8^–2.40	8.00 × 10^3^	8.00 × 10^4^	1.00 × 10^1^
N3-33	*tlh*	9.60 × 10^1^–9.60 × 10^−6^	1.92 × 10^0^	1.92 × 10^1^	1.00 × 10^1^	3.80 × 10^7^–3.80	1.27 × 10^2^	1.27 × 10^4^	1.00 × 10^2^
N4-9	*tlh*	9.72 × 10^1^–9.72 × 10^−6^	1.94 × 10^1^	1.94 × 10^2^	1.00 × 10^1^	9.40 × 10^7^–9.40	3.13 × 10^2^	3.13 × 10^5^	1.00 × 10^3^
N4-26	*tlh*	7.31 × 10^1^–7.31 × 10^−6^	1.46 × 10^0^	1.46 × 10^2^	1.00 × 10^2^	8.30 × 10^7^–8.30	2.77 × 10^3^	2.77 × 10^5^	1.00 × 10^2^
N4-31	*tlh*	9.37 × 10^1^–9.37 × 10^−6^	1.87 × 10^−4^	1.87 × 10^0^	1.00 × 10^4^	2.70 × 10^8^–2.70	9.00 × 10^3^	9.00 × 10^5^	1.00 × 10^2^
N4-46	*tlh*	8.34 × 10^1^–8.34 × 10^−6^	1.67 × 10^−2^	1.67 × 10^0^	1.00 × 10^2^	8.60 × 10^7^–8.60	2.87 × 10^3^	2.87 × 10^4^	1.00 × 10^1^
N5-15	*tlh*	7.26 × 10^1^–7.26 × 10^−6^	1.45 × 10^−3^	1.45 × 10^−1^	1.00 × 10^2^	1.30 × 10^8^–1.30	4.33 × 10^3^	4.33 × 10^4^	1.00 × 10^1^
N6-7	*tlh*	1.79 × 10^2^–1.79 × 10^−5^	3.58 × 10^−2^	3.58 × 10^0^	1.00 × 10^2^	1.61 × 10^8^–1.61	5.37 × 10^3^	5.37 × 10^4^	1.00 × 10^1^
N6- 10	*tlh*	1.21 × 10^2^–1.21 × 10^−5^	2.42 × 10^−3^	2.42 × 10^−1^	1.00 × 10^2^	1.41 × 10^8^–1.41	4.70 × 10^3^	4.70 × 10^5^	1.00 × 10^2^
N6-16	*tlh*	9.80 × 10^1^–9.80 × 10^−6^	1.96 × 10^1^	1.96 × 10^2^	1.00 × 10^1^	2.54 × 10^8^–2.40	8.47 × 10^2^	8.47 × 10^3^	1.00 × 10^1^
N6-26	*tlh*	1.20 × 10^2^–1.20 × 10^−5^	2.39 × 10^−1^	2.39 × 10^1^	1.00 × 10^2^	1.76 × 10^8^–1.76	5.87 × 10^3^	5.87 × 10^4^	1.00 × 10^1^
N7-3	*tlh*	8.79 × 10^1^–8.79 × 10^−6^	1.76 × 10^0^	1.76 × 10^1^	1.00 × 10^1^	9.50 × 10^7^–9.50	3.17 × 10^3^	3.17 × 10^5^	1.00 × 10^2^
N7-9	*tlh*	1.53 × 10^2^–1.53 × 10^−5^	3.05 × 10^−3^	3.05 × 10^−2^	1.00 × 10^1^	3.60 × 10^8^–3.60	1.20 × 10^3^	1.20 × 10^6^	1.00 × 10^3^
N7-45	*tlh*	1.22 × 10^2^–1.22 × 10^−5^	2.43 × 10^−1^	2.43 × 10^0^	1.00 × 10^1^	2.76 × 10^8^–2.76	9.20 × 10^2^	9.20 × 10^6^	1.00 × 10^4^
N7-19	*tlh*	1.54 × 10^2^–1.54 × 10^−5^	3.07 × 10^−1^	3.07 × 10^2^	1.00 × 10^3^	1.02 × 10^9^–1.02	3.40 × 10^2^	3.40 × 10^5^	1.00 × 10^3^
N8-9	*tlh*	1.09 × 10^2^–1.09 × 10^−5^	2.18 × 10^−1^	2.18 × 10^1^	1.00 × 10^2^	1.34 × 10^8^–1.34	4.47 × 10^3^	4.47 × 10^5^	1.00 × 10^2^
N8-13	*tlh*	1.82 × 10^2^–1.82 × 10^−5^	3.64 × 10^0^	3.64 × 10^2^	1.00 × 10^2^	2.45 × 10^8^–2.45	8.17 × 10^3^	8.17 × 10^5^	1.00 × 10^2^
N8-36	*tlh*	9.17 × 10^1^–9.17 × 10^−6^	1.83 × 10^−1^	1.83 × 10^2^	1.00 × 10^3^	2.18 × 10^8^–2.18	7.27 × 10^2^	7.27 × 10^5^	1.00 × 10^3^
N9-24	*tlh*	1.02 × 10^2^–1.02 × 10^−5^	2.05 × 10^−1^	2.05 × 10^1^	1.00 × 10^2^	3.10 × 10^7^–3.10	1.03 × 10^2^	1.03 × 10^5^	1.00 × 10^3^
N9-31	*tlh*	2.08 × 10^2^–2.08 × 10^−5^	4.16 × 10^−1^	4.16 × 10^2^	1.00 × 10^3^	2.60 × 10^8^–2.60	8.67 × 10^3^	8.67 × 10^4^	1.00 × 10^1^
N10-20	*tlh*	1.00 × 10^2^–1.00 × 10^−5^	2.00 × 10^0^	2.00 × 10^2^	1.00 × 10^2^	2.53 × 10^8^–2.53	8.43 × 10^3^	8.43 × 10^5^	1.00 × 10^2^
N10-48	*tlh*	1.16 × 10^2^–1.16 × 10^−5^	2.32 × 10^−1^	2.32 × 10^2^	1.00 × 10^3^	2.56 × 10^8^–2.56	8.53 × 10^2^	8.53 × 10^4^	1.00 × 10^2^
Q5-6	*tlh*	2.23 × 10^2^–2.23 × 10^−5^	4.46 × 10^−3^	4.46 × 10^0^	1.00 × 10^3^	1.68 × 10^8^–1.68	5.60 × 10^1^	5.60 × 10^3^	1.00 × 10^2^
Q8-7	*tlh*	1.89 × 10^2^–1.90 × 10^−5^	3.79 × 10^−2^	3.79 × 10^0^	1.00 × 10^2^	4.10 × 10^7^–4.10	1.37 × 10^0^	1.37 × 10^2^	1.00 × 10^2^
Q8-15	*tlh*	1.10 × 10^2^–1.10 × 10^−5^	2.21 × 10^−1^	2.21 × 10^1^	1.00 × 10^2^	2.37 × 10^8^–2.37	7.90 × 10^0^	7.90 × 10^1^	1.00 × 10^1^
ATCC 17802	*tlh*	9.26 × 10^1^–9.26 × 10^−6^	1.85 × 10^−4^	1.85 × 10^−1^	1.00 × 10^3^	1.32 × 10^8^–1.32	4.40 × 10^1^	4.40 × 10^3^	1.00 × 10^2^
ATCC 17802	*ureC*	9.26 × 10^1^–9.26 ×10^−6^	1.85 × 10^−3^	1.85 × 10^−^^1^	1.00 × 10^2^	1.32 × 10^8^–1.32	4.40 × 10^−1^	4.40 × 10^0^	1.00 × 10^1^

**Table 5 pathogens-11-00010-t005:** Sensitivity of the LAMP method and the PCR assay for the detection of aquatic product samples spiked with *V. parahaemolyticus* strains.

Target Gene	Aquatic Product	Spiked Strain	Cell Culture Dilutions (CFU/mL)	LOD (CFU/Reaction)		Rate of LOD (LAMP/PCR)
				LAMP	PCR	
*opaR*	*Aristichthys nobilis*	N7-19	2.96 × 10^8^–2.96	9.87 × 10^0^	9.87 × 10^1^	1.00 × 10^1^
	*Carassius auratus*			9.87 × 10^1^	9.87 × 10^2^	1.00 × 10^1^
	*Ctenopharyngodon idella*			9.87 × 10^0^	9.87 × 10^2^	1.00 × 10^2^
	*Parabramis pekinensis*			9.87 × 10^2^	9.87 × 10^4^	1.00 × 10^2^
	*Mytilus edulis*			9.87 × 10^2^	9.87 × 10^3^	1.00 × 10^1^
	*Litopenaeus vannamei*			9.87 × 10^−2^	9.87 × 10^3^	1.00 × 10^5^
*vpadF*	*Aristichthys nobilis*	N7-19	2.96 × 10^8^–2.96	9.87 × 10^0^	9.87 × 10^3^	1.00 × 10^3^
	*Carassius auratus*			9.87 × 10^−1^	9.87 × 10^2^	1.00 × 10^3^
	*Ctenopharyngodon idella*			9.87 × 10^2^	9.87 × 10^4^	1.00 × 10^2^
	*Parabramis pekinensis*			9.87 × 10^2^	9.87 × 10^4^	1.00 × 10^2^
	*Mytilus edulis*			9.87 × 10^2^	9.87 × 10^4^	1.00 × 10^2^
	*Litopenaeus vannamei*			9.87 × 10^0^	9.87 × 10^3^	1.00 × 10^3^
*tlh*	*Aristichthys nobilis*	N7-19	2.96 × 10^8^–2.96	9.87 × 10^3^	9.87 × 10^5^	1.00 × 10^2^
	*Carassius auratus*			9.87 × 10^2^	9.87 × 10^4^	1.00 × 10^2^
	*Ctenopharyngodon idella*			9.87 × 10^3^	9.87 × 10^4^	1.00 × 10^1^
	*Parabramis pekinensis*			9.87 × 10^3^	9.87 × 10^5^	1.00 × 10^2^
	*Mytilus edulis*			9.87 × 10^4^	9.87 × 10^5^	1.00 × 10^1^
	*Litopenaeus vannamei*			9.87 × 10^2^	9.87 × 10^3^	1.00 × 10^1^
*ureC*	*Aristichthys nobilis*	ATCC17802	2.75 × 10^9^–2.75	9.17 × 10^3^	9.17 × 10^5^	1.00 × 10^2^
	*Carassius auratus*			9.17 × 10^2^	9.17 × 10^4^	1.00 × 10^2^
	*Ctenopharyngodon idella*			9.17 × 10^3^	9.17 × 10^4^	1.00 × 10^1^
	*Parabramis pekinensis*			9.17 × 10^3^	9.17 × 10^5^	1.00 × 10^2^
	*Mytilus edulis*			9.17 × 10^1^	9.17 × 10^2^	1.00 × 10^1^
	*Litopenaeus vannamei*			9.17 × 10^2^	9.17 × 10^4^	1.00 × 10^2^

**Table 6 pathogens-11-00010-t006:** Detection of the virulence-related genes of *V. parahaemolyticus* in drinking water and aquatic product samples by the LAMP method.

Sample	No. of Sample	Virulence-Related Gene	No. of Sample
Water sample			
Mineral water	3	*opaR^-^/vpadF^-^/tlh^-^/ureC^-^*	3
Tap water	3	*opaR^-^/vpadF^-^/tlh^-^/ureC^-^*	3
River water	3	*opaR^-^/vpadF^-^/tlh^-^/ureC^-^*	3
Lake water	3	*opaR^-^/vpadF^-^/tlh^-^/ureC^-^*	3
Estuarine water	3	*opaR^-^/vpadF^-^/tlh^-^/ureC^-^*	3
Meat sample			
*Aristichthys nobilis*	3	*opaR^-^/vpadF^-^/tlh-/ureC^-^*	3
*Carassius auratus*	3	*opaR^-^/vpadF^-^/tlh^-^/ureC^-^*	3
*Ctenopharyngodon idella*	3	*opaR^-^/vpadF^-^/tlh^-^/ureC^-^*	3
*Parabramis pekinensis*	3	*opaR^-^/vpadF^-^/tlh^-^/ureC^-^*	3
*Mytilus edulis*	3	*opaR^-^/vpadF^-^/tlh^-^/ureC^-^*	3
*Litopenaeus vannamei*	3	*opaR^-^/vpadF^-^/tlh^-^/ureC^-^*	3
Intestine sample			
*Aristichthys nobilis*	3	*opaR^-^/vpadF^-^/tlh^-^/ureC^-^*	3
*Carassius auratus*	3	*opaR^-^/vpadF^-^/tlh^-^/ureC^-^*	3
*Ctenopharyngodon idella*	3	*opaR^-^/vpadF^-^/tlh^-^/ureC^-^*	3
*Parabramis pekinensis*	3	*opaR^+^/vpadF^-^/tlh^+^/ure C^-^*	3
*Mytilus edulis*	3	*opaR^-^/vpadF^-^/tlh^-^/ureC^-^*	3
*Litopenaeus vannamei*	3	*opaR^-^/vpadF^-^/tlh^-^/ureC^-^*	3

## Data Availability

The datasets generated during and/or analyzed during the current study can be find in the main text and the Appendix A.

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
