# Peer review of "Qualitative and Quantitative Detection of Potentially Virulent Vibrio parahaemolyticus in Drinking Water and Commonly Consumed Aquatic Products by Loop-Mediated Isothermal Amplification"

_pathogens, 2021, doi:10.3390/pathogens11010010_

Round 1

Reviewer 1 Report

The manuscript entitled 'Qualitative and Quantitative Detection of Potentially Virulent Vibrio parahaemolyticus in Drinking Water and  Commonly Consumed Aquatic Products by Loop-Mediated 4 Isothermal Amplification' presented a good study regarding the Quantitative Detection of Potentially Virulent Vibrio parahaemolyticus in Drinking Water and seafood. The study is well-designed. However, several important details is missing which might exclude the potential acceptance in Science of the Total Environment.

Detail comments:
line 80: 'current literature in this field for the opaR, tlh, vpadF and ureC genes is rare' why authors selected these genes for LAMP, which is critical and is missing
line 204: 2.3.2. For the detection of Genomic DNA of V. parahaemolyticus
It is better to elucidate DNA copies per reaction and average detection time.
line 300: 2.5. Sensitivity of the LAMP Method for the Detection of Spiked Fish, Shrimp and Shellfish 300 Samples
Are these seafoods are Vibrio parahaemolyticus-free before spiked with Vibrio parahaemolyticus?
line 361: 
the incidence of V. parahaemolyticus in sewage water is also very high. However, it remains unclear whether author's approach is also good for dealing with these matrix?

line 391:V. harvey should be V. harveyi
line 392: K. pneumonia should be K. pneumoniae
line 414: Moreover, our data revealed that the L. vannamei matrix appeared to interfere with the LAMP method more than the others from the fish and shellfish. 
Authors need to briefly discuss the reasons. 
line 427: the limitation of LAMP should also be mentioned.

Reviewer 2 Report

Summary

The authors focused on four genes related to the infection process of V. parahaemolyticus and developed a LAMP method targeting these genes. The specificity of the new method was examined in a total of 50 strains, 16 Vibrio spp. and 34 non-Vibrio spp. The sensitivity of the developed new method was compared with that of the conventional standard PCR method on dozens of cultured V. parahaemolyticusstrains each carrying the target gene and on marine products spiked with the bacteria.

Furthermore, the authors used this method to detect V. parahaemolyticus in water samples collected from the environment and in marine products, and succeeded in detecting the bacteria in the intestinal contents of marine products.

Comments

I respect the effort of analyzing a large number of samples, but the huge amount of data makes this manuscript difficult to read. Fig2 shows that the sensitivity varies depending on the strain, and the sensitivity is better than the PCR method. However, Fig3 and Fig4, which show exactly the same results with different target genes, should be described as "data not shown" or supplemental figures unless there is something special to discuss.

Is there any reason why we should focus on the four genes targeted in this study? The value of this manuscript will be greatly increased if it is described the problems of the LAMP method targeting tdh and trh, which are already available, and the advantages of the developed LAMP method over them.

There are many places where the unit of LOD is described as “CFU/reaction”, but what do you think about the fact that this value is after the decimal point? I understand that it is calculated that way, but does it make sense to say "The detection limit is 0.03 V. parahaemolyticus"? I think it would be easier to understand the data if it is considered by the total volume of the reaction solution or sample to show the minimum number of bacteria that can actually be detected.

L23-26

Instead of stating that "bacteria were detected" in the conclusion of the abstract, the usefulness of the developed method should be demonstrated by the fact that bacteria could be detected in actual samples.

L97-109

This section should be listed under Materials & Methods.

L379

The data of "routine microbial isolation and identification" should be showed. If it is difficult to show the results, it should at least be stated in the Materials and methods section what methods were used for isolation and identification.

table2-4

There are two “Rate of LOD”s listed. If it is read carefully, I can understand that the one on the left is based on genomic DNA and the one on the right is based on cell culture, but you should try to make it clear at a glance.

fig4

The names of bacteria in the legend should be written in italics.
